# HiGS: History-Guided Sampling for Plug-and-Play Enhancement of Diffusion Models

**Seyedmorteza Sadat[1], Farnood Salehi[2], Romann M. Weber[2]**
[1]ETH Zürich, [2]DisneyResearch|Studios
{seyedmorteza.sadat}@inf.ethz.ch
{farnood.salehi, romann.weber}@disneyresearch.com

## ABSTRACT

While diffusion models have made remarkable progress in image generation, their outputs can still appear unrealistic and lack fine details, especially when using fewer number of neural function evaluations (NFEs) or lower guidance scales. To address this issue, we propose a novel momentum-based sampling technique, termed history-guided sampling (HiGS), which enhances quality and efficiency of diffusion sampling by integrating recent model predictions into each inference step. Specifically, HiGS leverages the difference between the current prediction and a weighted average of past predictions to steer the sampling process toward more realistic outputs with better details and structure. Our approach introduces practically no additional computation and integrates seamlessly into existing diffusion frameworks, requiring neither extra training nor fine-tuning. Extensive experiments show that HiGS consistently improves image quality across diverse models and architectures and under varying sampling budgets and guidance scales. Moreover, using a pretrained SiT model, HiGS achieves a new state-of-the-art FID of 1.61 for unguided ImageNet generation at 256×256 with only 30 sampling steps (instead of the standard 250). We thus present HiGS as a plug-and-play enhancement to standard diffusion sampling that enables faster generation with higher fidelity.

|  CFG  |  + HiGS (Ours)  |
|-------|-----------------|

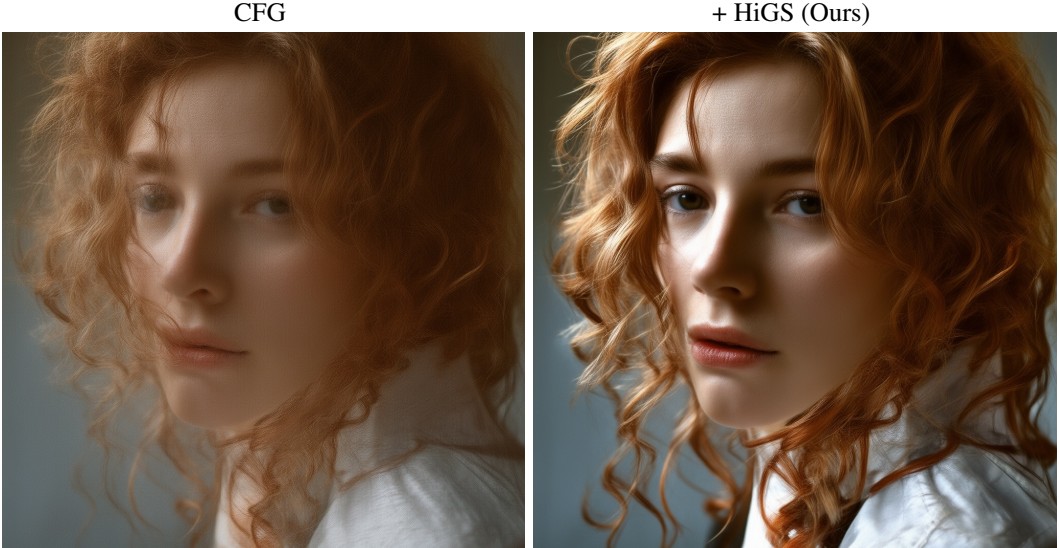

Portrait photo of a female with red hair

Figure 1: Sampling with diffusion models using fewer steps or lower guidance scales often results in blurry images with artifacts. We propose HiGS, a novel sampling method that enhances quality and details in generated images under various sampling budgets and guidance scales by leveraging a weighted average of the diffusion model's past predictions at each inference step. The results shown are generated with Stable Diffusion 3.5 (Esser et al., 2024) using 10 steps and a guidance scale of 1.2.

# 1 INTRODUCTION

Diffusion models (Sohl-Dickstein et al., 2015; Ho et al., 2020; Song et al., 2021b) are a class of generative models that learn the data distribution by reversing a forward noising process that gradually transforms data points into Gaussian noise. Although in theory, the reverse process should yield high-quality samples from the target distribution, diffusion models can produce outputs that are blurry or lack details due to optimization errors and inaccurate estimations of the data distribution at intermediate time steps. This issue becomes more pronounced when using fewer sampling steps or lower classifier-free guidance scales (Karras et al., 2022).

The generation process in diffusion models typically involves multiple neural function evaluations (NFEs), which are computationally expensive, especially for large-scale diffusion models containing billions of parameters (Esser et al., 2024). Reducing the number of NFEs reduces the sampling cost but leads to outputs that lack clarity, detail, and coherent global structures, as seen in Figure 1. Although various sampling methods have been proposed to reduce the required sampling steps (Lu et al., 2022; Karras et al., 2022), existing samplers still rely on relatively high step counts (e.g., 50) to achieve satisfactory results (Podell et al., 2023). Finding training-free methods that enable generating high-quality samples with fewer NFEs remains an open research question.

In addition to iterative sampling, modern diffusion models use high guidance scales to enhance sample quality and prompt alignment (Podell et al., 2023; Nichol et al., 2022). Classifier-free guidance (CFG) (Ho et al., 2021) has proven essential for reducing outliers and improving generation quality (Karras et al., 2024). However, CFG doubles the network forward passes per sampling step, thereby increasing the computational cost of inference, and higher guidance scales often lead to oversaturation and reduced diversity (Sadat et al., 2025a; 2024). Consequently, further research is needed to enhance generation quality when using lower guidance scales combined with fewer NFEs.

In this paper, We investigate the sampling process in diffusion models and propose a training-free, momentum-based modification of inference that consistently improves global structure, detail, and sharpness across different sampling budgets and guidance scales. Inspired by momentum-based variance reduction in stochastic optimization (Cutkosky & Orabona, 2019), we introduce history-guided sampling (HiGS), a novel method for improving the quality of diffusion models by leveraging the history of predictions made by the network. We demonstrate that this prediction history defines an effective guidance direction for steering the sampling trajectory toward higher-quality regions of the data distribution, especially under fewer NFEs or lower guidance scales. HiGS enables higher quality outputs and more efficient sampling from pretrained models while also improving the generation quality at lower guidance scales to avoid the drawbacks of high CFG scales.

HiGS introduces no additional overhead to the sampling process and can be seamlessly integrated into existing diffusion models and samplers without extra training. Through extensive experiments across a range of models and setups (including distilled diffusion models), we show that HiGS consistently improves sample quality, particularly in scenarios involving fewer sampling steps or lower guidance scales. Our results indicate that HiGS achieves higher quality metrics compared to standard sampling methods, establishing it as a universal enhancement for pretrained diffusion models under various sampling budgets and guidance scales. Furthermore, using a pretrained SiT model, HiGS achieves a state-of-the-art FID of 1.61 for unguided (i.e., without CFG) ImageNet generation at 256×256 while using only 30 sampling steps instead of the standard 250.

# 2 RELATED WORK

Score-based diffusion models (Song & Ermon, 2019; Song et al., 2021b; Sohl-Dickstein et al., 2015; Ho et al., 2020) learn complex data distributions by inverting a forward process that gradually adds Gaussian noise to the data. These models have rapidly advanced generative modeling, surpassing prior approaches in both fidelity and diversity (Nichol & Dhariwal, 2021; Dhariwal & Nichol, 2021). They have demonstrated state-of-the-art performance across a wide range of tasks, including unconditional and conditional image synthesis (Dhariwal & Nichol, 2021; Karras et al., 2022; Yu et al., 2025b; Karras et al., 2024), text-to-image generation (Podell et al., 2023; Esser et al., 2024; Labs, 2024; Qin et al., 2025), video generation (Blattmann et al., 2023b;a; Bar-Tal et al., 2024; Wan et al., 2025), image-to-image translation (Saharia et al., 2022a; Liu et al., 2023a; Xia et al., 2023), and audio generation (Chen et al., 2021; Huang et al., 2023; Liu et al., 2023b; Tian et al., 2025).

Since the introduction of DDPM (Ho et al., 2020), the field has seen substantial progress, with improvements spanning network architectures (Hoogeboom et al., 2023; Karras et al., 2023; Peebles & Xie, 2022; Dhariwal & Nichol, 2021), sampling techniques (Song et al., 2021a; Karras et al., 2022; Liu et al., 2022; Lu et al., 2022; Salimans & Ho, 2022), and training strategies (Nichol & Dhariwal, 2021; Karras et al., 2022; Song et al., 2021b; Salimans & Ho, 2022; Rombach et al., 2022). Despite these advancements, sampling from diffusion models still require relatively high step counts, and various guidance mechanisms—such as classifier guidance (Dhariwal & Nichol, 2021) and classifier-free guidance (Ho & Salimans, 2022)—remain critical for enhancing image quality and ensuring strong prompt alignment (Nichol et al., 2022).

A recent line of work has focused on developing better ODE solvers for the diffusion sampling process, often combined with improved training techniques to make the sampling ODE more linear (Lu et al., 2022; Karras et al., 2022; Esser et al., 2024). Despite these advances, most state-of-the-art samplers still require a relatively high number of sampling steps (e.g., 50 steps for Stable Diffusion XL (Podell et al., 2023)). Another direction of research explores distilling the diffusion model into a student network capable of sampling with fewer steps (Salimans & Ho, 2022; Song et al., 2023; Sauer et al., 2024). However, step distillation remains computationally expensive, requiring long training on advanced hardware. We show that HiGS serves as a training-free method to improve generation quality across various sampling budgets and networks (including distilled diffusion models).

Modern diffusion models often rely on high guidance scales to achieve strong image quality and prompt alignment. However, CFG doubles inference cost, and excessive CFG scales reduce diversity and cause oversaturation (Sadat et al., 2024; 2025a). On the other hand, sampling with lower CFG scales and NFEs avoid these issues but typically yield blurry images lacking fine detail and coherent structure. While several methods mitigate the drawbacks of high CFG scales (Kynkäänniemi et al., 2024; Sadat et al., 2025a; Wang et al., 2024), little attention has been given to improving quality under low CFG scales and limited sampling steps. We show that HiGS enhances generation across both low and high CFG regimes, offering benefits under varying CFG scales and sampling budgets.

In summary, sampling from diffusion models is an expensive iterative process that might produce unrealistic outputs under certain configurations. We propose a training-free method that improves generation quality across different sampling budgets and guidance scales, particularly in low-NFE and low-CFG scenarios. Thus, we present HiGS as a plug-and-play enhancement for diffusion sampling.

## 3 BACKGROUND

This section provides a brief overview of diffusion models. Let $\boldsymbol{x} \sim p_{\text{data}}(\boldsymbol{x})$ denote a data sample, and let $t \in [0, T]$ represent continuous time. The forward diffusion process gradually corrupts data by adding Gaussian noise: $\boldsymbol{z}_t = \boldsymbol{x} + \sigma(t)\boldsymbol{\epsilon}$, where $\boldsymbol{\epsilon} \sim \mathcal{N}(\mathbf{0}, \boldsymbol{I})$ and $\sigma(t)$ defines a monotonically increasing noise schedule, satisfying $\sigma(0) = 0$ and $\sigma(T) = \sigma_{\text{max}} \gg \sigma_{\text{data}}$. As shown by Karras et al. (2022), this forward process can be described by the following ordinary differential equation (ODE):

$$\mathrm{d}\boldsymbol{z} = -\dot{\sigma}(t)\sigma(t) \nabla_{\boldsymbol{z}_t} \log p_t(\boldsymbol{z}_t)\mathrm{d}t = -t \nabla_{\boldsymbol{z}_t} \log p_t(\boldsymbol{z}_t)\mathrm{d}t, \tag{1}$$

where we choose $\sigma(t) = t$, and $p_t(\boldsymbol{z}_t)$ is the distribution of noisy data at time $t$, with $p_0 = p_{\text{data}}$ and $p_T = \mathcal{N}(\mathbf{0}, \sigma_{\text{max}}^2 \boldsymbol{I})$. If the time-dependent score function $\nabla_{\boldsymbol{z}_t} \log p_t(\boldsymbol{z}_t)$ is known, one can sample from $p_{\text{data}}$ by integrating the ODE (or its stochastic counterpart) in reverse, from $t = T$ to $t = 0$.

In practice, the score function is approximated using a neural denoiser $D_{\boldsymbol{\theta}}(\boldsymbol{z}_t, t)$, trained to recover the clean data $\boldsymbol{x}$ from the noisy input $\boldsymbol{z}_t$. Conditional generation is supported by extending the denoiser to take an additional input $\boldsymbol{y}$ (e.g., class labels or text), resulting in $D_{\boldsymbol{\theta}}(\boldsymbol{z}_t, t, \boldsymbol{y})$.

**Classifier-free guidance (CFG)** Classifier-free guidance (CFG) is a technique for enhancing sample quality during inference by interpolating between unconditional and conditional model predictions (Ho & Salimans, 2022). Given an unconditional prediction $D_{\boldsymbol{\theta}}(\boldsymbol{z}_t, t)$, the guided denoiser output at each sampling step is computed as:

$$D_{\text{CFG}}(\boldsymbol{z}_t, t, \boldsymbol{y}) = w_{\text{CFG}} D_{\boldsymbol{\theta}}(\boldsymbol{z}_t, t, \boldsymbol{y}) - (w_{\text{CFG}} - 1) D_{\boldsymbol{\theta}}(\boldsymbol{z}_t, t), \tag{2}$$

where $w_{\text{CFG}} = 1$ corresponds to no guidance, and larger values increase conditioning strength. The unconditional model is typically trained by randomly dropping the condition $\boldsymbol{y}$ with some probability $p \in [0.1, 0.2]$ during training. Alternatively, a separate network or the conditional model itself can

be used to estimate the unconditional score (Karras et al., 2023; Sadat et al., 2025b). While CFG is known to improve perceptual quality, it often comes at the cost of increased oversaturation and reduced sample diversity (Sadat et al., 2025a; 2024).

## 4 SAMPLING WITH PREDICTION HISTORY

Let $t_0 > t_1 > \ldots > t_M$ be the sampling time grid with $M+1$ steps. The model prediction at time step $t_k$ is denoted by $D_{\boldsymbol{\theta}}(\boldsymbol{z}_{t_k}, t_k, \boldsymbol{y})$. Given a window of size $W \geq 1$, we define the history $\mathcal{H}_k$ at step $k$ as the set of past predictions prior to $t_k$, i.e., $\mathcal{H}_k := \{D_{\boldsymbol{\theta}}(\boldsymbol{z}_{t_i}, t_i, \boldsymbol{y})\}_{i \in I_k}$ for $I_k := \{\max(0, k - W), \ldots, k-1\}$. We next show how leveraging this history enhances sampling quality. For simplicity, we define $D_c(\boldsymbol{z}_{t_k}) \doteq D_{\boldsymbol{\theta}}(\boldsymbol{z}_{t_k}, t_k, \boldsymbol{y})$, $D_u(\boldsymbol{z}_{t_k}) \doteq D_{\boldsymbol{\theta}}(\boldsymbol{z}_{t_k}, t_k)$, and $D_{\mathrm{CFG}}(\boldsymbol{z}_{t_k}) \doteq D_{\mathrm{CFG}}(\boldsymbol{z}_{t_k}, t_k, \boldsymbol{y})$ to represent the conditional, unconditional, and CFG outputs, respectively.

### 4.1 MOTIVATION

We first claim that the Euler sampler for diffusion models can be interpreted as performing stochastic gradient descent (SGD) on a time-varying energy function. To show this, consider the ODE in Equation (1) and a discretization $\{t_0, t_1, \ldots, t_M\}$. A single Euler step at time $t_k$ can be written as

$$\boldsymbol{z}_{t_{k+1}} = \boldsymbol{z}_{t_k} + t_k(t_k - t_{k+1})\nabla_{\boldsymbol{z}_{t_k}} \log p_{t_k}(\boldsymbol{z}_{t_k}) \tag{3}$$

$$= \boldsymbol{z}_{t_k} - t_k(t_k - t_{k+1})\nabla_{\boldsymbol{z}_{t_k}} E_{t_k}(\boldsymbol{z}_{t_k}), \tag{4}$$

where the energy function $E_t(\boldsymbol{z}_t)$ is defined via $p_t(\boldsymbol{z}_t) = \frac{1}{Z}\exp(-E_t(\boldsymbol{z}_t))$ for some normalization constant $Z$. This shows that each Euler step in diffusion sampling corresponds to a step of gradient descent on the time-dependent energy $E_t(\boldsymbol{z}_t)$ with learning rate $t_k(t_k - t_{k+1})$.

Motivated by this new perspective, we argue that we can enhance this gradient estimate to improve the efficiency and quality of the sampling process in diffusion models. One promising approach is to augment the gradient with an additional momentum-based term similar to STORM (Cutkosky & Orabona, 2019), a variance-reduction method for non-convex optimization. Specifically, given a differentiable function $f$, the momentum term in STORM is defined as $\nabla f(\boldsymbol{z}_{t_k}) - \nabla f(\boldsymbol{z}_{t_{k-1}})$, which incorporates information from consecutive steps for more stable updates. Applying this to Equation (3), we obtain $\nabla E_{t_k}(\boldsymbol{z}_{t_k}) - \nabla E_{t_{k-1}}(\boldsymbol{z}_{t_{k-1}})$ as the momentum term. Equivalently, this can be seen as incorporating the residual of past score terms or model outputs, since the score function corresponds to the energy gradient. In the following, we further generalize this idea to incorporate multiple past predictions rather than only the previous step (similar to multistep ODE solvers (Atkinson et al., 2009)). An alternative perspective is given in Appendix A, where we connect classifier-free guidance to gradient ascent on a specific objective. In Appendix B, we show that the history terms reduce the Euler solver's local truncation error from $\mathcal{O}(h_k^2)$ to $\mathcal{O}(h_k^3)$, where $h_k = t_k - t_{k+1}$, thereby improving the global error from $\mathcal{O}(h)$ to $\mathcal{O}(h^2)$ with $h = \max_k h_k$.

### 4.2 HISTORY-GUIDED SAMPLING

Building on the insights discussed above, we propose a novel sampling method for diffusion models, termed history-guided sampling (HiGS), which integrates past predictions into the current sampling step. We leverage the set of past predictions $\mathcal{H}_k$ at each sampling step $t_k$ to improve generation quality such as sharpness, details and global structure. We detail each step of HiGS below.

**Buffer input** When the sampling is done with CFG, we can choose to keep track of the CFG-guided predictions $D_{\mathrm{CFG}}(\boldsymbol{z}_{t_k})$ or the conditional outputs $D_c(\boldsymbol{z}_{t_k})$ as the history. We found that the CFG-guided predictions are more effective in improving the quality of sampling. Thus, we use $D_{\mathrm{CFG}}(\boldsymbol{z}_{t_k})$ as the inputs to $\mathcal{H}_k$ when CFG is enabled, leading to $\mathcal{H}_k := \{D_{\mathrm{CFG}}(\boldsymbol{z}_{t_i})\}_{i \in I_k}$.

**Incorporating the history** The next design choice is how to use $\mathcal{H}_k$ during sampling. Specifically, we want a function $g$ that determines the influence of past predictions on the current step. By generalizing the momentum update rule in STORM, we adopt an EMA-style weighted average that emphasizes recent predictions:

$$g(\mathcal{H}_k) = \sum_{i \in I_k} \alpha(1-\alpha)^{k-1-i} D_{\mathrm{CFG}}(\boldsymbol{z}_{t_i}), \tag{5}$$

where $\alpha \in (0, 1)$ is a hyperparameter. This formulation integrates information from history while prioritizing recent steps for more informative guidance. We later show that several alternative definitions of $g$, such as simple averaging, are viable options (see Appendix E for more details).

Let $\Delta D_{t_k} = D_{\text{CFG}}(\boldsymbol{z}_{t_k}) - g(\mathcal{H}_k)$ denote the guidance term in HiGS. A straightforward strategy for using $\Delta D_{t_k}$ at inference is to combine this update with the current output, analogous to CFG, via $D_{\text{CFG}}(\boldsymbol{z}_{t_k}) + w_{\text{HiGS}}\Delta D_{t_k}$ with scale $w_{\text{HiGS}}$. In the following, we introduce several improvements over this naive baseline that we found crucial to substantially boost performance. Appendix E presents extensive ablation studies to support various design choices of HiGS.

**Scheduling the guidance weight**    In our experiments, the benefits of HiGS were most evident during the early and middle sampling steps. We noticed that as sampling progresses, the update term introduces diminishing improvements and may even cause noisy artifacts. To address this, we use a weight schedule $w_{\text{HiGS}}(t_k)$ that adapts the scale of the guidance term according to the time step $t_k$. We employ a square-root schedule given by

$$w_{\text{HiGS}}(t) = \begin{cases} 0 & t \leq t_{\min}, \\ w_{\text{HiGS}}\sqrt{\frac{t-t_{\min}}{t_{\max}-t_{\min}}} & t_{\min} < t \leq t_{\max}, \\ 0 & t > t_{\max}. \end{cases} \tag{6}$$

**Optional orthogonal projection**    Additionally, we found that it is sometimes beneficial to project the update vector $\Delta D_{t_k}$ on $D_{\text{CFG}}(\boldsymbol{z}_{t_k})$ and downweight its parallel component to prevent oversaturation and color artifacts (especially at higher values of $w_{\text{HiGS}}$) (Sadat et al., 2025a). The parallel component in $\Delta D_{t_k}$ can be computed via

$$\Delta D_{t_k}^{\|} = \frac{\langle \Delta D_{t_k}, D_{\text{CFG}}(\boldsymbol{z}_{t_k}) \rangle}{\langle D_{\text{CFG}}(\boldsymbol{z}_{t_k}), D_{\text{CFG}}(\boldsymbol{z}_{t_k}) \rangle} D_{\text{CFG}}(\boldsymbol{z}_{t_k}), \tag{7}$$

and the orthogonal component is computed via $\Delta D_{t_k}^{\perp} = D_{\text{CFG}}(\boldsymbol{z}_{t_k}) - \Delta D_{t_k}^{\|}$. We use $\Delta D_{t_k}(\eta) = \Delta D_{t_k}^{\perp} + \eta \Delta D_{t_k}^{\|}$ as the projected update direction, where $\eta \in [0, 1]$ is a hyperparameter. As demonstrated in Figure 11 (appendix), the orthogonal projection step can mitigate oversaturation and color artifacts in the generated outputs.

**Frequency-domain filtering**    We further observed that using $\Delta D_{t_k}(\eta)$ as the guidance term generally leads to unrealistic color compositions in generations (see Figure 15 in the appendix). To solve this, we employ frequency-based high-pass filtering using the discrete cosine transform (DCT). Since color composition corresponds to low-frequency contents of an image, we apply DCT to the update term $\Delta D_{t_k}(\eta)$ and attenuate its low-frequency signals with a sigmoid high-pass filter:

$$H(R) = \text{Sigmoid}(\lambda(R - R_c)), \tag{8}$$

where $R = \sqrt{f_x^2 + f_y^2}$ is the radial frequency at coordinates $(x, y)$, $R_c$ is the cutoff threshold, and $\lambda$ controls the sharpness of the transition. This procedure effectively removes the color shifts, leading to more realistic and visually consistent outputs. Accordingly, the final update rule for HiGS is

$$D_{\text{HiGS}}(\boldsymbol{z}_{t_k}) = D_{\text{CFG}}(\boldsymbol{z}_{t_k}) + w_{\text{HiGS}}(t_k) \, \text{iDCT}\big(H(R) \cdot \text{DCT}(\Delta D_{t_k}(\eta))\big). \tag{9}$$

## 5    EXPERIMENTS

We now provide extensive qualitative and quantitative comparisons between standard sampling and sampling with HiGS to show that HiGS enhances the performance of diffusion models under various setups. Further experiments, ablations, and implementation details are provided in the appendix.

**Setup**    We evaluate our method primarily on text-to-image generation using Stable Diffusion models (Rombach et al., 2022; Podell et al., 2023; Esser et al., 2024), and on class-conditional generation with ImageNet (Russakovsky et al., 2015) using DiT-XL/2 (Peebles & Xie, 2022), and SiT-XL + REPA (Yu et al., 2025a; Leng et al., 2025). For each model, we employ the default sampling algorithms (e.g., the Euler solver for Stable Diffusion XL) and rely on the official pretrained checkpoints and publicly released codebases to ensure alignment with the original implementations. Additional details regarding the experimental configurations and hyperparameters are provided in Appendix F.

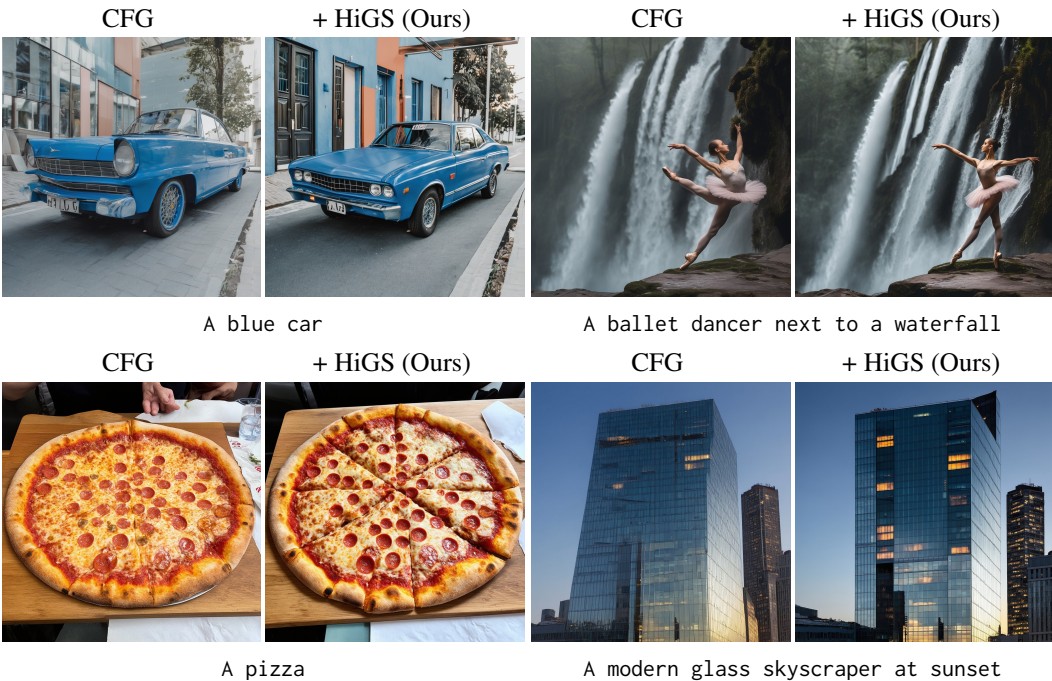

Figure 2: Effect of HiGS on generated samples using standard sampling setups. Compared with CFG outputs, HiGS produces sharper images with improved structure and fewer artifacts.

**Evaluation metrics** For text-to-image generation, we primarily use HPSv2 (Wu et al., 2023) as our main quality and prompt alignment metric, as we found it to be the most aligned with human judgment. We also report ImageReward (Xu et al., 2023) and the HPSv2 win rate as additional preference scores, along with the CLIP Score (Hessel et al., 2021) for completeness. For class-conditional models, we use the Fréchet Inception Distance (FID) (Heusel et al., 2017) as the primary metric to assess both image fidelity and diversity, given its strong alignment with human visual perception in this setting. To ensure fair comparisons, all FID evaluations are conducted under a standardized setup to minimize sensitivity to implementation differences. We also report Inception Score (IS) (Salimans et al., 2016), Precision (Kynkäänniemi et al., 2019), and Recall (Kynkäänniemi et al., 2019) as complementary metrics to separately evaluate sample quality and diversity.

## 5.1 MAIN RESULTS

**Qualitative comparisons** We first qualitatively evaluate the impact of HiGS on generation quality under three different regimes: the normal setup using practical CFG scales with high NFEs (Figure 2), sampling with fewer number of NFEs (Figure 3), and sampling under low CFG scales (Figure 4). We note that HiGS is able to improve generation quality under all these settings, showing that its benefits cover a wide range of CFG scales and sampling budgets.

**HiGS vs CFG scale** Next, we quantitatively evaluate the effect of adding HiGS to the sampling process on generation quality across different CFG scales $w_{\text{CFG}}$. Figure 5a shows that HiGS is beneficial across all guidance scales, maintaining a significant margin in terms of HPS win rate.

**HiGS vs the number of sampling steps** Similarly, Figure 5b shows that HiGS outperforms the CFG baseline in generation quality across all NFEs. This demonstrates that HiGS is beneficial for all sampling budgets, and its advantages are not limited to a specific number of sampling steps.

**Comparison with different models** Furthermore, we evaluate the impact of HiGS on output quality across different models and metrics in Tables 1 and 2, using a fixed guidance scale and sampling steps per baseline for both sampling with and without HiGS to ensure a fair comparison. As before, HiGS consistently improves generation quality across diverse setups and metrics, indicating that it serves as a universal enhancement to standard diffusion sampling.

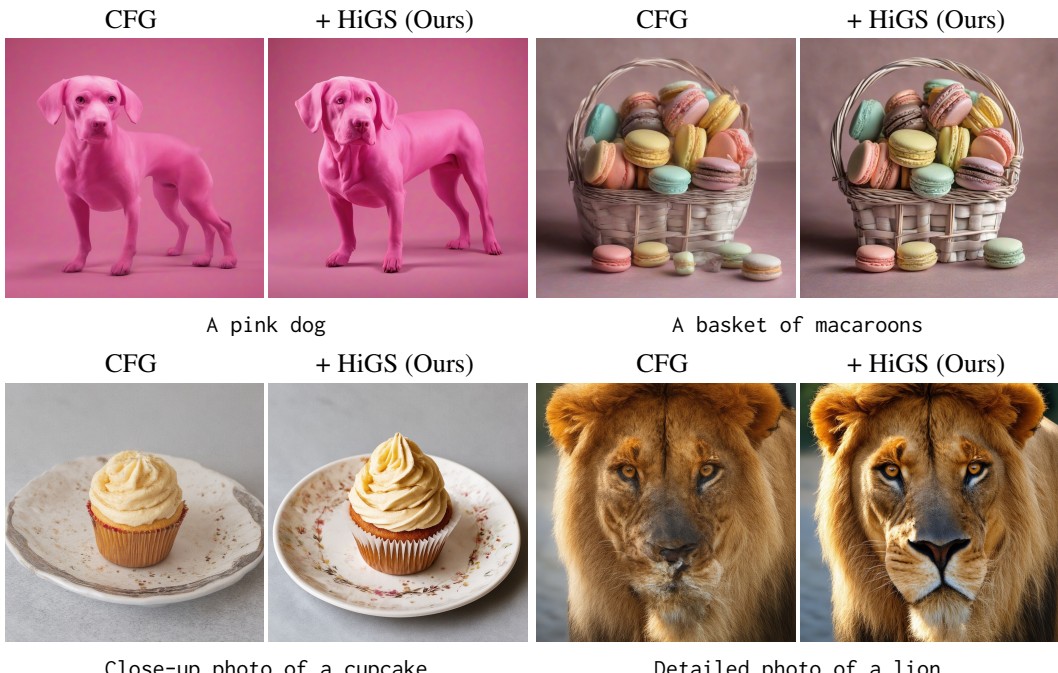

Figure 3: Comparison of standard sampling and HiGS under fewer number NFEs. HiGS outputs exhibit significantly better global structure, sharpness, and details than the baseline.

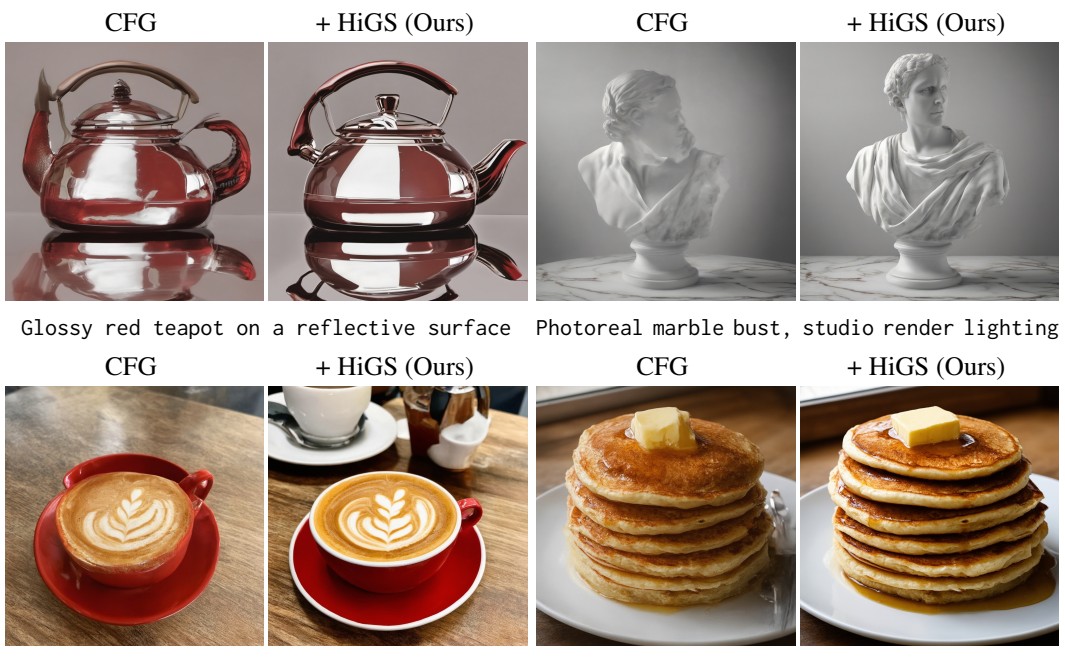

Figure 4: Illustration of the effect of HiGS on generated outputs under lower guidance scales. HiGS leads to noticeable improvements in image quality and details compared to the baseline.

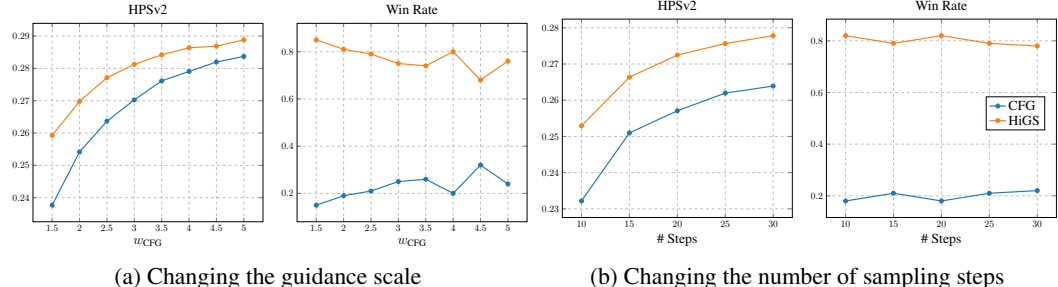

(a) Changing the guidance scale  (b) Changing the number of sampling steps

Figure 5: Effect of HiGS across guidance scales and sampling budgets using Stable Diffusion 3 (Esser et al., 2024). HiGS consistently outperforms standard sampling with CFG in all settings, highlighting its effectiveness in improving generation quality under varying guidance scales and sampling budgets.

Table 1: Quantitative evaluation of the effect of HiGS on sampling. HiGS consistently improves all metrics across different models, demonstrating higher generation quality than the CFG baseline. For fairness, sampling with and without HiGS is performed using the same NFE and CFG scale.

| Model | Guidance | FID ↓ | IS ↑ | Precision ↑ | Recall ↑ |
|---|---|---|---|---|---|
| SiT-XL + REPA (Yu et al., 2025a) | CFG | 12.08 | 187.11 | 0.68 | **0.73** |
| | HiGS (Ours) | **4.86** | **277.20** | **0.80** | 0.70 |
| DiT-XL/2 (Peebles & Xie, 2022) | CFG | 8.73 | 173.21 | 0.72 | 0.68 |
| | HiGS (Ours) | **7.15** | **180.05** | **0.75** | **0.71** |
| Stable Diffusion XL (Podell et al., 2023) | CFG | 28.49 | 35.07 | 0.56 | 0.54 |
| | +HiGS (Ours) | **26.18** | **36.22** | **0.59** | **0.57** |
| Stable Diffusion 3 (Esser et al., 2024) | CFG | 27.19 | 40.11 | 0.73 | 0.41 |
| | +HiGS (Ours) | **26.84** | **40.94** | **0.76** | **0.42** |

Table 2: Evaluation of using HiGS with various Stable Diffusion models. HiGS improves all metrics reflecting human preference for image quality and prompt alignment across multiple benchmarks. All comparisons are conducted with the same NFE and CFG scale for fairness.

| Benchmark | Model | Guidance | ImageReward ↑ | HPSv2 ↑ | Win Rate ↑ |
|---|---|---|---|---|---|
| DrawBench (Saharia et al., 2022b) | Stable Diffusion XL | CFG | -0.091 | 0.224 | 0.07 |
| | | +HiGS (Ours) | **0.148** | **0.249** | **0.93** |
| | Stable Diffusion 3 | CFG | 0.491 | 0.257 | 0.18 |
| | | +HiGS (Ours) | **0.621** | **0.272** | **0.82** |
| | Stable Diffusion 3.5 | CFG | 0.621 | 0.258 | 0.21 |
| | | +HiGS (Ours) | **0.702** | **0.270** | **0.79** |
| Parti Prompts (Yu et al., 2022) | Stable Diffusion XL | CFG | 0.191 | 0.239 | 0.08 |
| | | +HiGS (Ours) | **0.360** | **0.261** | **0.92** |
| | Stable Diffusion 3 | CFG | 0.843 | 0.273 | 0.19 |
| | | +HiGS (Ours) | **0.919** | **0.285** | **0.81** |
| | Stable Diffusion 3.5 | CFG | 0.879 | 0.270 | 0.18 |
| | | +HiGS (Ours) | **0.935** | **0.282** | **0.82** |
| HPS Prompts (Wu et al., 2023) | Stable Diffusion XL | CFG | 0.327 | 0.245 | 0.04 |
| | | +HiGS (Ours) | **0.515** | **0.275** | **0.96** |
| | Stable Diffusion 3 | CFG | 0.820 | 0.279 | 0.21 |
| | | +HiGS (Ours) | **0.901** | **0.291** | **0.79** |
| | Stable Diffusion 3.5 | CFG | 0.821 | 0.274 | 0.18 |
| | | +HiGS (Ours) | **0.889** | **0.289** | **0.82** |

## 5.2 IMAGENET BENCHMARK

Table 3 shows that HiGS improves the performance of recent state-of-the-art models on conditional ImageNet generation at 256×256 resolution. For unguided generation (i.e., without CFG), HiGS

Table 3: Effect of adding HiGS to recent state-of-the-art methods for conditional ImageNet generation at 256×256. HiGS significantly improves sampling speed, matching the FID of the original models in just 30–40 steps. Moreover, HiGS achieves a new state-of-the-art FID of 1.61 for conditional generation without CFG using the SiT-XL + REPA-E model (Leng et al., 2025).

| Model | Guidance | # Steps ↓ | FID ↓ | IS ↑ | Precision ↑ | Recall ↑ |
|---|---|---|---|---|---|---|
| REPA-E (Leng et al., 2025) | Unguided | 250 | 1.83 | 217.30 | 0.77 | **0.66** |
| | +HiGS (Ours) | **30** | **1.61** | **240.75** | **0.81** | 0.62 |
| | CFG | 250 | 1.26 | **314.90** | 0.79 | **0.66** |
| | +HiGS (Ours) | **40** | 1.32 | 306.10 | **0.80** | 0.65 |
| REPA (Yu et al., 2025a) | Unguided | 250 | 5.90 | 157.80 | 0.70 | **0.69** |
| | +HiGS (Ours) | **40** | **5.43** | **165.91** | **0.71** | 0.68 |
| | CFG | 250 | **1.42** | 305.70 | 0.80 | **0.65** |
| | +HiGS (Ours) | **40** | 1.44 | **306.80** | **0.80** | 0.64 |

Table 4: Effect of adding HiGS to the sampling process of distilled models. HiGS improves the quality of these models, showing that its effects are complementary to diffusion distillation.

| Model | Guidance | ImageReward ↑ | HPSv2 ↑ | Win Rate ↑ | CLIP Score ↑ |
|---|---|---|---|---|---|
| SDXL-Flash (SD-Community) | CFG | 0.774 | 0.273 | 0.03 | 0.332 |
| | +HiGS (Ours) | **0.864** | **0.298** | **0.97** | **0.333** |
| SDXL-Lightning (Lin et al., 2024) | CFG | 0.63 | 0.277 | 0.18 | **0.318** |
| | +HiGS (Ours) | **0.66** | **0.285** | **0.82** | 0.317 |

reduces the state-of-the-art FID from 1.83 to 1.61 using only 30 sampling steps, without any retraining of the base model. Moreover, by improving sample quality at lower NFEs, HiGS provides a training-free way to accelerate sampling from existing models. For guided generation with CFG, HiGS matches the performance of the base models in just 40 sampling steps, compared to 250 steps with the default sampler. Finally, this experiment shows that HiGS is compatible with guidance interval (Kynkäänniemi et al., 2024), as both CFG baselines incorporate this technique in their samplers.

## 5.3 COMPATIBILITY WITH DISTILLED MODELS

Finally, we also demonstrate that HiGS is compatible with distilled diffusion models that use fewer sampling steps. Diffusion distillation reduces sampling steps by training a student model to replicate the performance of the base model. As shown in Table 4, HiGS enhances the sampling quality of these distilled models, achieving a significant win rate according to the HPS score. This shows that the benefits of HiGS are complementary to diffusion distillation, and that HiGS can be applied as a training-free method to further enhance the quality of distilled diffusion models.

## 6 DISCUSSION AND CONCLUSION

In this work, we presented history-guided sampling (HiGS), a simple, training-free modification of diffusion sampling that leverages the history of model predictions to steer the reverse process toward higher quality and more coherent images. By applying a history-informed correction that emphasizes the deviation between the current prediction and a weighted average of past predictions, HiGS mitigates blur and artifacts that often arise during sampling, particularly with fewer number of NFEs or lower CFG scales. HiGS requires no additional network evaluations, integrates seamlessly with existing samplers and architectures, and is fully plug-and-play. Our experiments demonstrated that HiGS consistently improves perceptual quality and fidelity across diverse models and sampling budgets, with especially strong benefits in low-NFE and low-CFG regimes. While highly effective, HiGS still inherits, albeit to a lesser extent, the biases and some limitations of the underlying diffusion models, and addressing these challenges remains a promising direction for future work.

BROADER IMPACT STATEMENT

Our method has the potential to improve the realism and quality of outputs produced by diffusion models without requiring expensive retraining, making it practically valuable for applications in visual content creation. However, as generative modeling technologies continue to evolve, the ease of producing and distributing synthetic or misleading content also increases. While such advances can significantly boost creativity and productivity, they also raise important ethical and societal concerns. It is therefore essential to raise awareness about the potential misuse of generative models and to carefully consider the broader societal implications of their deployment. For an in-depth discussion on ethics and creativity in generative modeling, we refer readers to Lin & Losavio (2025).

REPRODUCIBILITY STATEMENT

Our work builds upon the official implementations of the pretrained models cited in the main text. Algorithm 1 gives the algorithm for applying HiGS at inference, and the pseudocode for HiGS is presented in Algorithms 2 and 3. Additional implementation details, including the hyperparameters used in the main experiments, are given in Appendix F.

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

## A CFG AS GRADIENT ASCENT

This section provides an alternative intuition behind HiGS. Recall that CFG update at time step $t_k$ can be written in the alternate form

$$D_{\text{CFG}}(\boldsymbol{z}_{t_k}) = D_c(\boldsymbol{z}_{t_k}) + (w_{\text{CFG}} - 1)(D_c(\boldsymbol{z}_{t_k}) - D_u(\boldsymbol{z}_{t_k})), \tag{10}$$

which can be interpreted as a single gradient ascent step on the objective

$$f_{\text{CFG}}(D_c(\boldsymbol{z}_{t_k}), D_u(\boldsymbol{z}_{t_k})) = \frac{1}{2} \| D_c(\boldsymbol{z}_{t_k}) - \text{sg}[D_u(\boldsymbol{z}_{t_k})] \|^2 \tag{11}$$

w.r.t. the model output $D_c(\boldsymbol{z}_{t_k})$ (Sadat et al., 2025a). Here, $\text{sg}[\cdot]$ denotes the stop-gradient operation. This suggests that we can augment the gradient update in CFG with a history-based momentum term similar to STORM (Cutkosky & Orabona, 2019). For a differentiable function $f$, STORM uses the gradient difference $\nabla f(\boldsymbol{z}_{t_k}) - \nabla f(\boldsymbol{z}_{t_{k-1}})$ to incorporate information from consecutive steps. Applying this to the CFG objective from Equation (11), we have

$$\nabla f_{\text{CFG}}(\boldsymbol{z}_{t_k}) = D_c(\boldsymbol{z}_{t_k}) - D_u(\boldsymbol{z}_{t_k}), \tag{12}$$
$$\nabla f_{\text{CFG}}(\boldsymbol{z}_{t_{k-1}}) = D_c(\boldsymbol{z}_{t_{k-1}}) - D_u(\boldsymbol{z}_{t_k}), \tag{13}$$

so the STORM momentum reduces to

$$\nabla f_{\text{CFG}}(\boldsymbol{z}_{t_k}) - \nabla f_{\text{CFG}}(\boldsymbol{z}_{t_{k-1}}) = D_c(\boldsymbol{z}_{t_k}) - D_c(\boldsymbol{z}_{t_{k-1}}). \tag{14}$$

This suggests enhancing the guidance direction with a history term along $D_c(\boldsymbol{z}_{t_k}) - D_c(\boldsymbol{z}_{t_{k-1}})$. In Section 4, we generalized this idea to incorporate multiple past predictions rather than only $D_c(\boldsymbol{z}_{t_{k-1}})$ and introduced several refinements that further improve generation quality.

## B ERROR ANALYSIS OF HiGS

We now show that adding HiGS to the Euler solver for diffusion models can improve the convergence rate of its local truncation error, and hence leading to better global estimates with fewer sampling steps. Let $u(\boldsymbol{z}_t, t) = t \nabla_{\boldsymbol{z}_t} \log p_t(\boldsymbol{z}_t)$. The sampling ODE for diffusion models is then equal to $d\boldsymbol{z} = -u(\boldsymbol{z}_t, t)dt$. The Euler step for this ODE at time step $t_k$ is equal to

$$\hat{\boldsymbol{z}}_{t_{k+1}} = \hat{\boldsymbol{z}}_{t_k} + (t_k - t_{k+1})u(\hat{\boldsymbol{z}}_{t_k}, t_k) \tag{15}$$
$$= \hat{\boldsymbol{z}}_{t_k} + h_k u(\hat{\boldsymbol{z}}_{t_k}, t_k), \tag{16}$$

where $h_k = t_k - t_{k+1}$ is the step size, and $\hat{\boldsymbol{z}}_{t_0} \sim \mathcal{N}(\boldsymbol{0}, \sigma_{\max}^2)$. Now, assume that instead of $u(\boldsymbol{z}_{t_k}, t_k)$, we follow an update similar to HiGS based on the previous update $u(\boldsymbol{z}_{t_{k-1}}, t_{k-1})$, i.e., we use

$$\tilde{u}(\boldsymbol{z}_{t_k}, t_k) = u(\boldsymbol{z}_{t_k}, t_k) + w_k(u(\boldsymbol{z}_{t_k}, t_k) - u(\boldsymbol{z}_{t_{k-1}}, t_{k-1})) \tag{17}$$

for some time dependent weight schedule $w_k = w(t_k)$.

**Theorem B.1.** *Let $u(\boldsymbol{z}, t)$ be sufficiently smooth in both arguments with bounded derivatives, and let $h_k = t_k - t_{k+1}$ denote the variable step size for a time grid $t_0 > t_1 > \ldots > t_M$ with $M + 1$ steps. Then, at each time step $t_k$, the Euler update in Equation (15) has local truncation error $\mathcal{O}(h_k^2)$. In contrast, there exists a weight $w_k \in \mathbb{R}$ such that the modified update rule of Equation (17) used in HiGS, yields local truncation error $\mathcal{O}(h_k^3)$. Consequently, the global error improves from $\mathcal{O}(h)$ for Euler to $\mathcal{O}(h^2)$ with HiGS, where $h = \max_k(t_k - t_{k+1})$.*

*Proof.* Let $\boldsymbol{z}(t)$ denote the ground truth solution to the sampling ODE (derived by integration). For simplicity, we define $\boldsymbol{z}_k = \boldsymbol{z}(t_k)$. Using Taylor expansion, we get

$$\boldsymbol{z}_{k+1} = \boldsymbol{z}_k + \frac{d\boldsymbol{z}}{dt}(t_k)(t_{k+1} - t_k) + \frac{1}{2}\frac{d^2\boldsymbol{z}}{dt^2}(\zeta_1)(t_{k+1} - t_k)^2 \tag{18}$$

for some $\zeta_1 \in [t_{k+1}, t_k]$. Since we have $\frac{d\boldsymbol{z}}{dt} = -u(\boldsymbol{z}, t)$, we get

$$\boldsymbol{z}_{k+1} = \boldsymbol{z}_k - (t_{k+1} - t_k)u(\boldsymbol{z}_k, t_k) + \frac{1}{2}\frac{d^2\boldsymbol{z}}{dt^2}(\zeta_1)(t_{k+1} - t_k)^2 \tag{19}$$

$$= \boldsymbol{z}_k + h_k u(\boldsymbol{z}_k, t_k) - \frac{h_k^2}{2}\frac{du}{dt}(\boldsymbol{z}(\zeta_1), \zeta_1). \tag{20}$$

Accordingly, the local truncation error $L_k$ of the euler method is given by

$$L_k = \|z_{k+1} - (z_k + h_k u(z_k, t_k))\| = \frac{h_k^2}{2} \left\| \frac{\mathrm{d}u}{\mathrm{d}t}(z(\zeta_1), \zeta_1) \right\| \le C h_k^2 \tag{21}$$

for some constant $C$. This shows that the local truncation error for the Euler solver is equal to $\mathcal{O}(h_k^2)$. We next show that using the history-based update in Equation (17) improves this rate to $\mathcal{O}(h_k^3)$. If we use the Taylor expansion of $u(z_{k-1}, t_{k-1})$, we get

$$u(z_{k-1}, t_{k-1}) = u(z_k, t_k) + h_{k-1}\frac{\mathrm{d}u}{\mathrm{d}t}(z_k, t_k) + \frac{h_{k-1}^2}{2}\frac{\mathrm{d}^2 u}{\mathrm{d}t^2}(z(\zeta_2), \zeta_2) \tag{22}$$

for some $\zeta_2 \in [t_k, t_{k-1}]$. This leads to

$$\tilde{u}(z_k, t_k) = u(z_k, t_k) - w_k h_{k-1}\frac{\mathrm{d}u}{\mathrm{d}t}(z_k, t_k) - \frac{w_k h_{k-1}^2}{2}\frac{\mathrm{d}^2 u}{\mathrm{d}t^2}(z(\zeta_2)\zeta_2). \tag{23}$$

Using another Taylor expansion for $z(t)$ (this time up to third order), we get

$$z_{k+1} = z_k + \frac{\mathrm{d}z}{\mathrm{d}t}(t_k)(t_{k+1} - t_k) + \frac{1}{2}\frac{\mathrm{d}^2 z}{\mathrm{d}t^2}(t_k)(t_{k+1} - t_k)^2 + \frac{1}{6}\frac{\mathrm{d}^3 z}{\mathrm{d}t^3}(\zeta_3)(t_{k+1} - t_k)^3 \tag{24}$$

$$= z_k + h_k u(z_k, t_k) - \frac{h_k^2}{2}\frac{\mathrm{d}u}{\mathrm{d}t}(z_k, t_k) + \frac{h_k^3}{6}\frac{\mathrm{d}^2 u}{\mathrm{d}t^2}(z(\zeta_3), \zeta_3) \tag{25}$$

for some $\zeta_3 \in [t_{k+1}, t_k]$. Accordingly, the new truncation error $\tilde{L}_k$ is given by

$$\tilde{L}_k = \|z_{k+1} - (z_k + h_k \tilde{u}(z_k, t_k))\| \tag{26}$$

$$= \left\| h_k(w_k h_{k-1} - \frac{h_k}{2})\frac{\mathrm{d}u}{\mathrm{d}t}(z_k, t_k) + \frac{h_k^3}{6}\frac{\mathrm{d}^2 u}{\mathrm{d}t^2}(z(\zeta_3), \zeta_3) + \frac{w_k h_k h_{k-1}^2}{2}\frac{\mathrm{d}^2 u}{\mathrm{d}t^2}(z(\zeta_2), \zeta_2) \right\|. \tag{27}$$

Now, if we set $w_k = \frac{h_k}{2h_{k-1}}$, the first term vanishes, and we get

$$\tilde{L}_k = \left\| \frac{h_k^3}{6}\frac{\mathrm{d}^2 u}{\mathrm{d}t^2}(z(\zeta_3), \zeta_3) + \frac{h_k^2 h_{k-1}}{4}\frac{\mathrm{d}^2 u}{\mathrm{d}t^2}(z(\zeta_2), \zeta_2) \right\|. \tag{28}$$

Since the step sizes are bounded, there is a constant $A$ such that $h_{k-1} \le A h_k$. We therefore have

$$\tilde{L}_k \le \frac{h_k^3}{6}\left\| \frac{\mathrm{d}^2 u}{\mathrm{d}t^2}(z(\zeta_3)\zeta_3) \right\| + \frac{A h_k^3}{4}\left\| \frac{\mathrm{d}^2 u}{\mathrm{d}t^2}(z(\zeta_2), \zeta_2) \right\|. \tag{29}$$

Assuming that $u(z, t)$ is sufficiently smooth, its derivatives should be bounded on $[t_M, t_0]$. Thus,

$$\tilde{L}_k \le C' h_k^3 = \mathcal{O}(h_k^3) \tag{30}$$

for some constant $C'$. Therefore, Theorem B.2 implies that for $h = \max_k(t_k - t_{k+1})$, the global error is $\mathcal{O}(h)$ for the Euler update and $\mathcal{O}(h^2)$ for the history-based update in HiGS. $\square$

**Theorem B.2.** *Let $u(z, t)$ be sufficiently smooth in both arguments, and consider a decreasing time grid $t_0 > t_1 > \cdots > t_M$ with variable step sizes $h_k = t_k - t_{k+1}$. Suppose a numerical ODE solver produces updates with local truncation error $L_k$ of order $\mathcal{O}(h_k^p)$, for some $p \ge 1$. Then the final global error of the solver $E_{t_0}$ satisfies $E_{t_0} = \mathcal{O}(h^p)$ for $h = \max_k(t_k - t_{k+1})$.*

*Proof.* A general proof is given in Chapter II.3, Theorem 3.4 of Hairer et al. (1993). $\square$

## C    RELATION TO AUTOGUIDANCE

Autoguidance (Karras et al., 2024) leverages a weaker diffusion model (a smaller or less trained variant of the base model) to improve generation quality. However, since it requires training an additional model, it cannot be applied out of the box to enhance pretrained models without extra training. In contrast, the history term introduced by HiGS can be interpreted as an explicit negative signal derived from the previous predictions of the diffusion model. Specifically, the prediction $D_c(z_{t_k})$ at each step corresponds to a denoised version of the current input $z_{t_k}$, which initially tends to be blurry and of lower quality. As sampling progresses, successive predictions become sharper, and thus the update term $D_c(z_{t_k}) - g(\mathcal{H}_k)$ roughly captures the difference between the base model's prediction and that of a "worse" version. Because the negative signal $g(\mathcal{H}_k)$ is defined entirely from past predictions, HiGS can be applied to pretrained models without any additional training. Moreover, HiGS can be combined with autoguidance by replacing the CFG prediction $D_{\text{CFG}}(z_{t_k})$ in HiGS with the corresponding prediction from autoguidance.

| APG | + HiGS (Ours) | APG | + HiGS (Ours) |

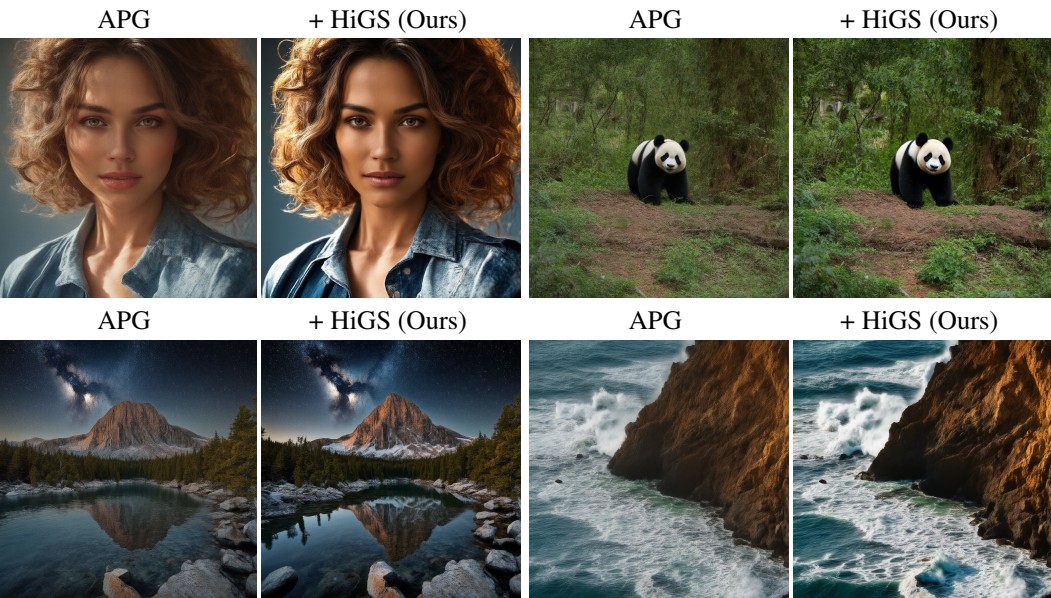

| APG | + HiGS (Ours) | APG | + HiGS (Ours) |

Figure 6: Compatibility of HiGS with APG (Sadat et al., 2025a) on Stable Diffusion 3 (Esser et al., 2024). HiGS enhances the quality and detail of APG, indicating that its benefits are complementary to various guidance methods.

## D ADDITIONAL EXPERIMENTS

**Sampling time**    Since HiGS does not require an additional forward pass, its sampling time is effectively identical to the CFG baseline. To verify this, we measured inference performance on Stable Diffusion 3 using the same GPU, with and without HiGS. In both cases, sampling achieved about 6.50 iterations per second with identical memory usage, confirming that HiGS has negligible impact on runtime or memory.

**Compatibility with adaptive projected guidance**    We next demonstrate that HiGS is compatible with other CFG variants such as adaptive projected guidance (APG) (Sadat et al., 2025a). Figure 6 shows that HiGS also improves the quality and details of the APG samples. This indicates that our approach is complementary to various CFG modifications.

**CLIP scores**    While HPSv2 and ImageReward both account for image quality and prompt alignment, we also report CLIP scores in this section for completeness. Table 5 shows that HiGS achieves the same CLIP score as the baseline, indicating that it preserves the prompt alignment of CFG. We note, however, that CLIP scores may not fully reflect human judgment, as they often show limited variation across models (Podell et al., 2023). Therefore, we primarily relied on HPSv2 as our main metric for evaluating both quality and prompt alignment.

**Compatibility with different samplers**    The main results of this paper examined the effect of adding HiGS to various models with their default samplers. In this experiment, we explicitly demonstrate that HiGS is compatible with different diffusion sampling techniques using DiT-XL/2 as the base model. Table 6 shows that HiGS improves the quality across all samplers (including multistep solvers such as DPM++), indicating that its benefits are complementary to changing the sampling method.

**Avoiding the issues of high CFG scales**    While increasing CFG generally improves image quality at various NFEs, high CFG scales often result in oversaturation and reduced diversity. In contrast, HiGS enhances image quality at lower CFG scales while inherently avoiding these drawbacks. Figures 7 and 8 demonstrate that applying HiGS at lower CFG scales yields more diverse and realistic generations compared to sampling with high CFG. Furthermore, as shown in the main paper, HiGS consistently outperforms CFG across a wide range of scales.

Table 5: Comparison of CLIP scores across datasets and models. HiGS consistently improves generation quality (see Table 2) while maintaining virtually identical CLIP scores. This demonstrates that the quality gains are achieved without sacrificing prompt alignment.

| Benchmark | Model | Guidance | CLIP Score ↑ |
|---|---|---|---|
| DrawBench (Saharia et al., 2022b) | Stable Diffusion XL | CFG | 0.308 |
| | | +HiGS (Ours) | **0.309** |
| | Stable Diffusion 3 | CFG | 0.329 |
| | | +HiGS (Ours) | 0.327 |
| | Stable Diffusion 3.5 | CFG | **0.334** |
| | | +HiGS (Ours) | 0.330 |
| Parti Prompts (Yu et al., 2022) | Stable Diffusion XL | CFG | 0.317 |
| | | +HiGS (Ours) | **0.318** |
| | Stable Diffusion 3 | CFG | **0.327** |
| | | +HiGS (Ours) | 0.325 |
| | Stable Diffusion 3.5 | CFG | **0.330** |
| | | +HiGS (Ours) | 0.329 |
| HPS Prompts (Wu et al., 2023) | Stable Diffusion XL | CFG | 0.333 |
| | | +HiGS (Ours) | **0.336** |
| | Stable Diffusion 3 | CFG | **0.332** |
| | | +HiGS (Ours) | 0.330 |
| | Stable Diffusion 3.5 | CFG | **0.337** |
| | | +HiGS (Ours) | 0.336 |

Table 6: Effect of adding HiGS to various popular diffusion samplers using the DiT-XL/2 model with 15 steps and $w_{\text{CFG}} = 1.25$. Note that HiGS improves the performance of all samplers (including multistep solvers such as DPM++), and hence its effect is complementary to changing the sampler.

| Sampler | Guidance | FID ↓ | IS ↑ | Precision ↑ | Recall ↑ |
|---|---|---|---|---|---|
| DDIM (Song et al., 2021a) | CFG | 11.87 | 151.40 | 0.69 | **0.70** |
| | +HiGS (Ours) | **8.73** | **173.21** | **0.73** | 0.69 |
| DPM++ (Lu et al., 2022) | CFG | 7.15 | 180.05 | 0.75 | **0.71** |
| | +HiGS (Ours) | **6.66** | **191.45** | **0.76** | 0.70 |
| DDPM (Ho et al., 2020) | CFG | 24.65 | 107.25 | 0.58 | **0.67** |
| | +HiGS (Ours) | **17.03** | **128.38** | **0.64** | 0.64 |
| PLMS (Liu et al., 2022) | CFG | 6.75 | 183.11 | 0.74 | **0.72** |
| | +HiGS (Ours) | **6.13** | **191.44** | **0.75** | 0.72 |
| UniPC (Zhao et al., 2023) | CFG | 6.79 | 185.86 | 0.75 | **0.71** |
| | +HiGS (Ours) | **6.60** | **192.81** | **0.76** | 0.69 |

**Changing the scale in HiGS** Figure 9 shows the performance of HiGS as the scale $w_{\text{HiGS}}$ varies. We observe that performance improves as $w_{\text{HiGS}}$ increases, but degrades when the scale becomes too large, while all settings still outperform the CFG baseline. Overall, we find that $w_{\text{HiGS}} \leq 3$ provides consistently strong performance across models.

## E  ABLATION STUDIES

**The choice for buffer input** Table 7 shows that both the conditional prediction $D_c(z_t)$ and the guided prediction $D_{\text{CFG}}(z_t)$ are viable inputs to HiGS, but using $D_{\text{CFG}}(z_t)$ as the history leads to better performance. Accordingly, we used $D_{\text{CFG}}(z_t)$ in our experiments whenever possible.

High CFG

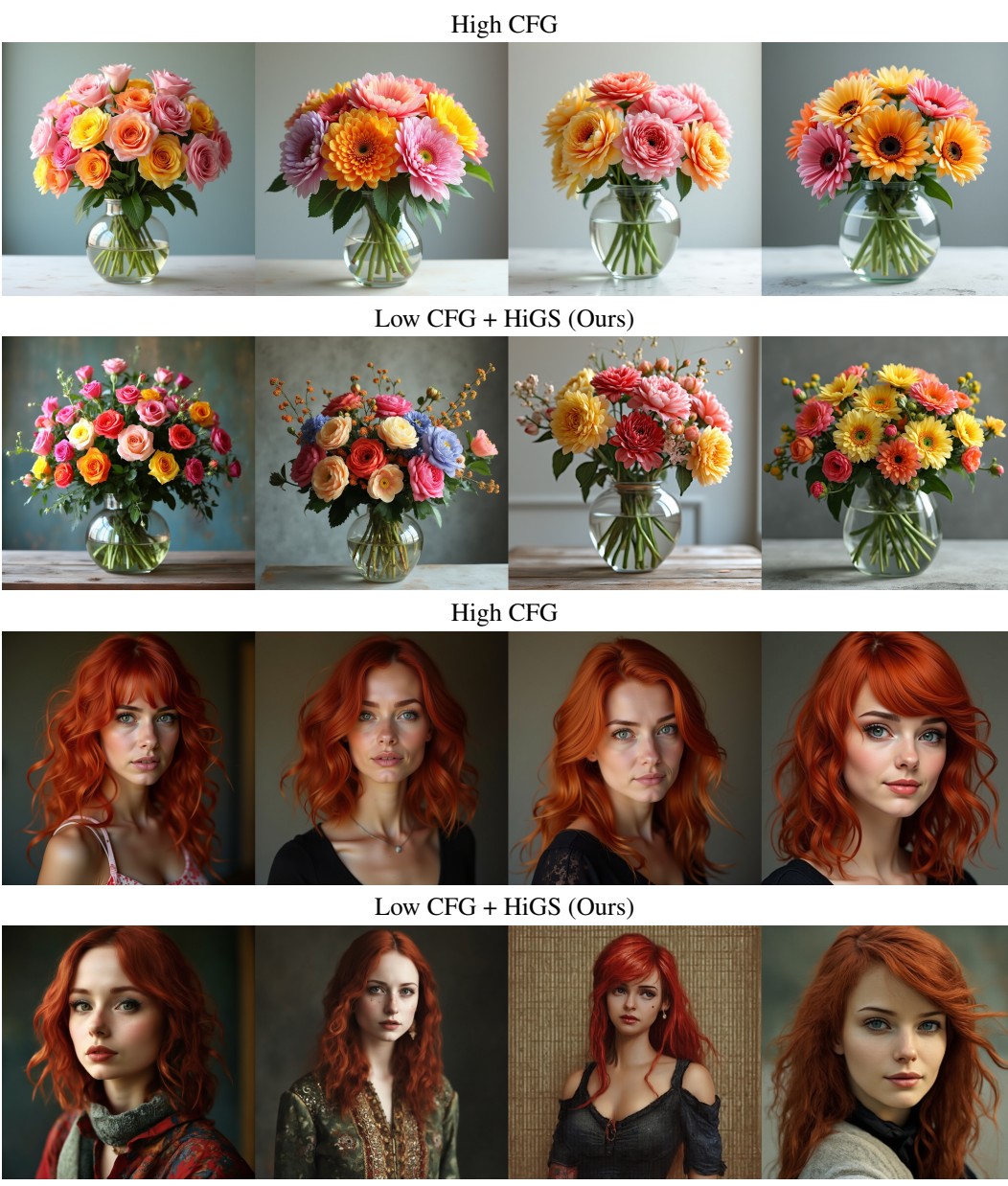

Low CFG + HiGS (Ours)

High CFG

Low CFG + HiGS (Ours)

Figure 7: High CFG scales improve overall structure of the image but lead to lack of diversity, oversaturation, and unrealistic effects. HiGS significantly improves the quality of lower CFG scales, leading to more realistic and diverse generations. Samples are generated with Flux (Labs, 2024).

**Frequency filtering** Figure 10 shows that applying frequency filtering is essential for avoiding color artifacts in our method. Without this step (both with and without projection), the images often exhibit unrealistic patterns and unnatural color compositions. These issues are effectively mitigated through our DCT-based filtering, and we later show that performance remains fairly robust to different choices of hyperparameters used in the DCT filter.

**Projection** We next demonstrate that projection can also reduce color artifacts in the generated images in some situations. Figure 11 shows that after projection, the generations appear more realistic with fewer oversaturated regions. Thus, incorporating projection into HiGS may lead to more realistic outputs when oversaturation exists.

High CFG

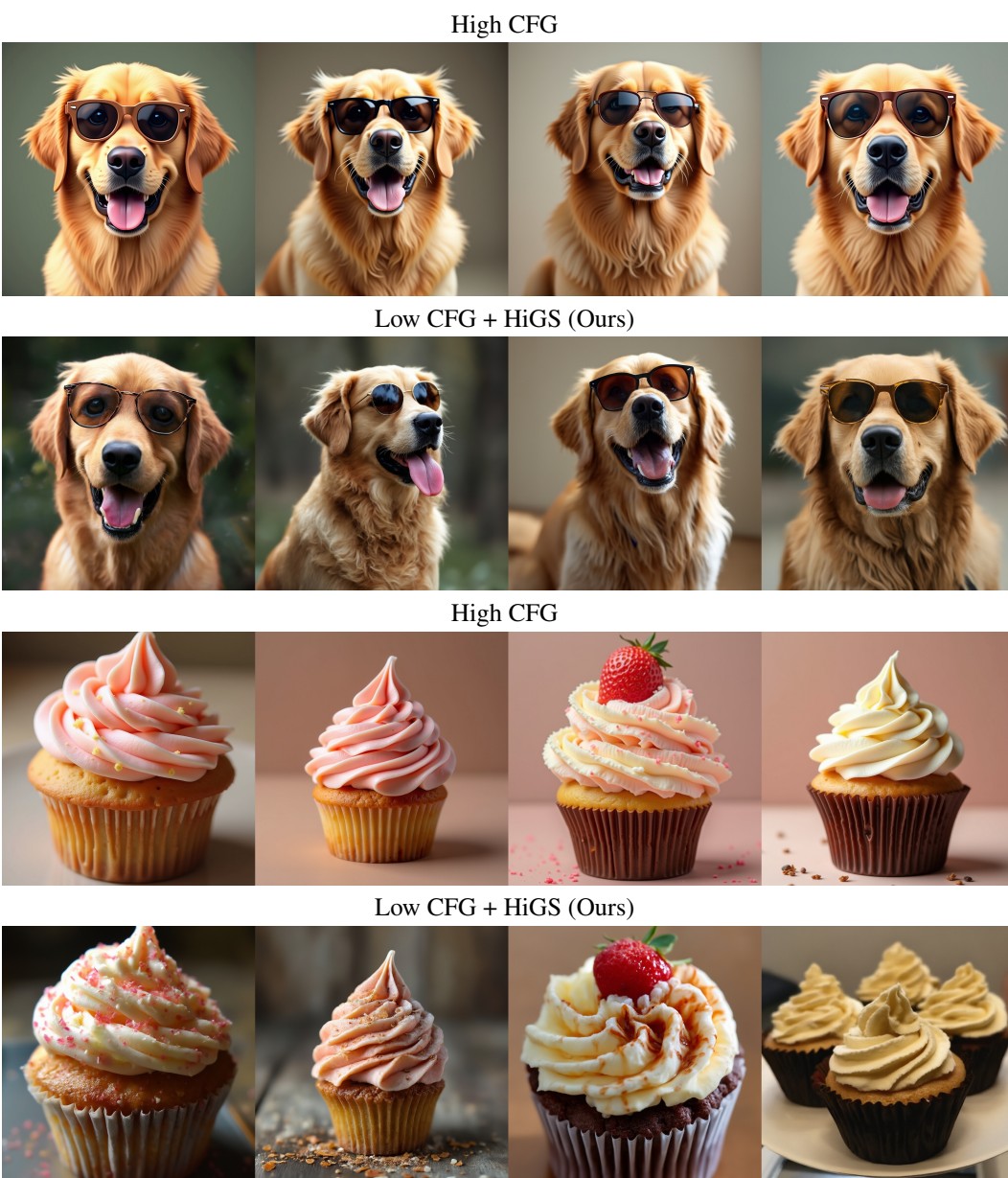

Low CFG + HiGS (Ours)

High CFG

Low CFG + HiGS (Ours)

Figure 8: High CFG scales improve overall structure of the image but lead to lack of diversity, oversaturation, and unrealistic effects. HiGS significantly improves the quality of lower CFG scales, leading to more realistic and diverse generations. Samples are generated with Flux (Labs, 2024).

**Effect of the thresholds in the weight schedule** We next analyze the impact of $t_{\min}$ and $t_{\max}$ in the weight scheduler. Figure 12 shows that setting $t_{\min}$ too low or too high leads to either excessive or insufficient guidance, resulting in suboptimal performance. Similarly, Figure 13 indicates that reducing $t_{\max}$ generally causes insufficient guidance and degraded results. Overall, we found that setting $t_{\min} \in [0.3, 0.5]$ and $t_{\max} \in [0.9, 1.0]$ yields consistently good performance across models.

**Effect of EMA value** Figure 14 shows that HiGS performs well for a wide range of EMA values $\alpha$. We have found that setting $\alpha = 0.5$ or $\alpha = 0.75$ gives good results across all models and architectures.

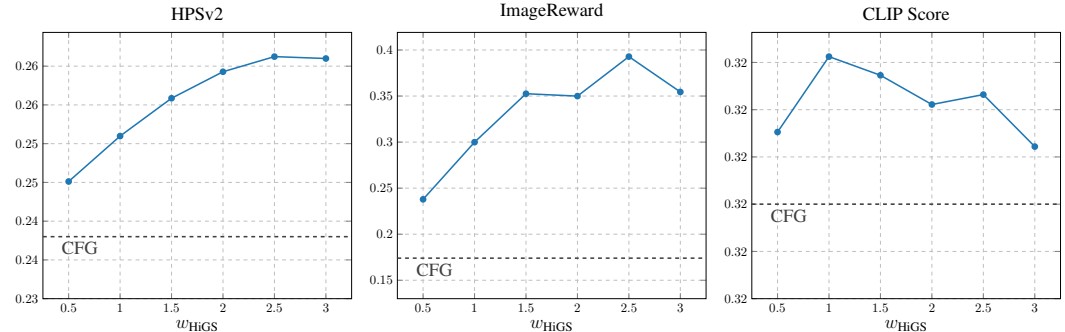

Figure 9: Effect of varying $w_{\text{HiGS}}$ on different quality metrics with Stable Diffusion XL (Podell et al., 2023). HiGS consistently outperforms the CFG baseline across a wide range of $w_{\text{HiGS}}$ scales. In practice, we found that setting $w_{\text{HiGS}} \leq 3$ generally leads to good performance across models.

Table 7: Comparison of using the conditional model prediction vs the CFG prediction as input to HiGS. While both options outperform baseline sampling without HiGS, using the CFG-guided prediction (when available) leads to better results.

| Config | HPSv2 ↑ | Image Reward ↑ | CLIP Score ↑ |
|---|---|---|---|
| Baseline (with CFG) | 0.238 | 0.174 | 0.317 |
| +HiGS (Conditional) | 0.249 | 0.234 | 0.315 |
| +HiGS (CFG) | **0.255** | **0.371** | **0.322** |

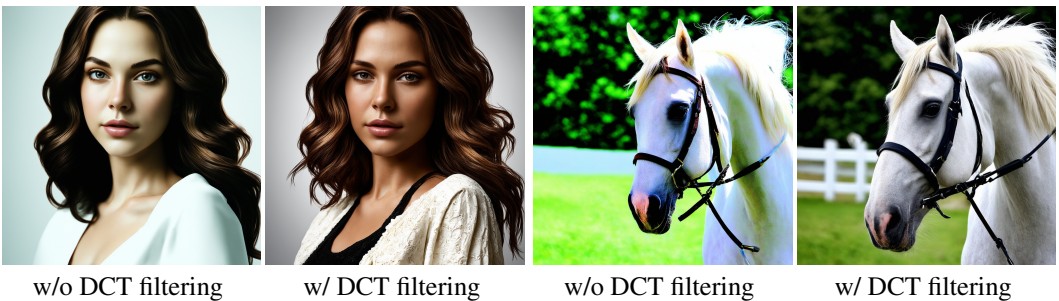

| w/o DCT filtering | w/ DCT filtering | w/o DCT filtering | w/ DCT filtering |

Figure 10: Effect of frequency filtering on generation quality. Without DCT filtering, the results show unrealistic color compositions, which are corrected after applying the filtering operation.

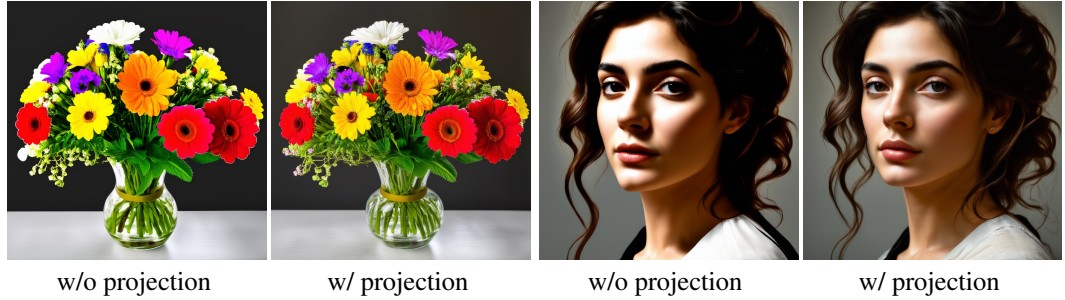

| w/o projection | w/ projection | w/o projection | w/ projection |

Figure 11: Effect of using projection in HiGS. Without projection, the generations might produce oversaturated results, which can be mitigated by reducing the strength of the parallel component.

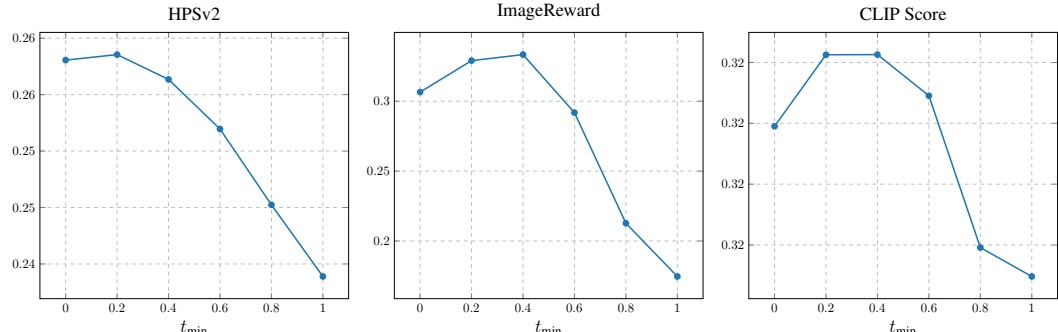

Figure 12: Effect of varying $t_{min}$ on different quality metrics for Stable Diffusion XL. The results show that performance degrades when $t_{min}$ is set too low or too high. We find that $t_{min} \in [0.3, 0.5]$ yields good results across all models.

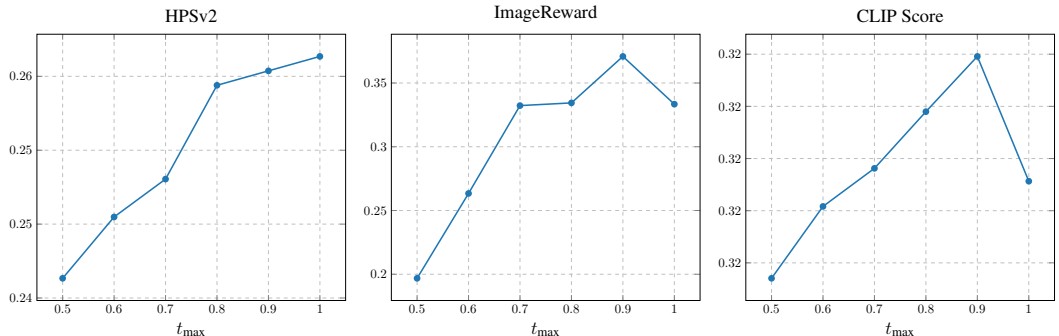

Figure 13: Effect of varying $t_{max}$ on different quality metrics for Stable Diffusion XL. The results show that reducing $t_{max}$ often leads to insufficient guidance and degraded quality. We recommend setting $t_{max} \in [0.9, 1]$ for all models.

**Effect of the DCT threshold** We show in this section that HiGS is relatively robust to the choice of the DCT threshold $R_c$ as long as it is sufficiently low. Figure 15 shows that the metrics are more or less the same for lower $R_c$ values. We set $R_c \approx 0.05$ for all models.

**Effect of weight scheduling** We next show that alternative choices for the weight scheduler are possible. Specifically, we test a constant function for $w_{HiGS}(t)$ applied over an interval, i.e.,

$$w_{HiGS}(t) = \begin{cases} 0 & t \le t_{min}, \\ w_{HiGS} & t_{min} < t \le t_{max}, \\ 0 & t > t_{max}, \end{cases} \tag{31}$$

as well as a linear schedule given by

$$w_{HiGS}(t) = \begin{cases} 0 & t \le t_{min}, \\ w_{HiGS} \cdot \frac{t - t_{min}}{t_{max} - t_{min}} & t_{min} < t \le t_{max}, \\ 0 & t > t_{max}. \end{cases} \tag{32}$$

Table 8 shows that all three options are viable. We chose the square-root function in Section 4, as we empirically found that it produces more visually appealing results.

**Various options for the history function** In Table 9, we show that different choices of $g$ yield similar quality metrics, indicating that HiGS is robust to this design decision. We adopt the EMA option, as it produced slightly more realistic outputs in our visual evaluations. Moreover, it enables computing the average on the fly without storing all past predictions in memory (see Algorithm 2).

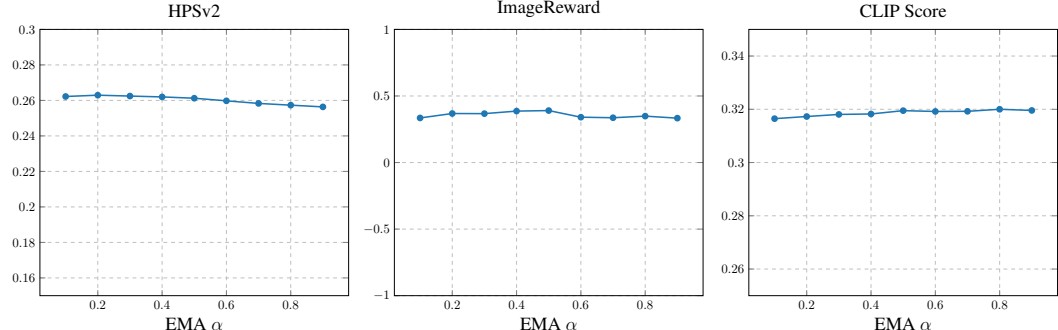

Figure 14: Effect of varying the EMA parameter $\alpha$ on different quality metrics for Stable Diffusion XL. The results indicate that performance remains consistent across choices of $\alpha$.

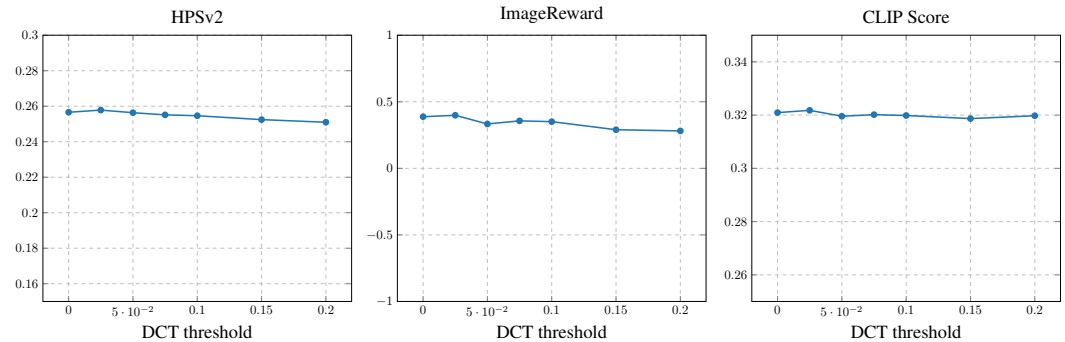

Figure 15: Effect of varying the DCT threshold on different quality metrics for Stable Diffusion XL. The results indicate that performance remains consistent across different threshold choices.

Table 8: Effect of different weight schedulers on HiGS.

| Weight Scheduler | $w_{\text{HiGS}}$ | HPSv2 ↑ | Image Reward ↑ | CLIP Score ↑ |
|---|---|---|---|---|
| Constant | 1.75 | **0.261** | 0.36 | **0.319** |
| Square-root | 2.50 | **0.261** | **0.39** | **0.319** |
| Linear | 3.25 | 0.260 | 0.37 | 0.318 |

Table 9: Effect of different averaging functions $g(\mathcal{H}_k)$ on HiGS.

| $g(\mathcal{H}_k)$ | HPSv2 ↑ | Image Reward ↑ | CLIP Score ↑ |
|---|---|---|---|
| Random | **0.261** | 0.362 | 0.318 |
| Average | **0.261** | 0.349 | 0.319 |
| Weighted average | 0.260 | 0.370 | 0.320 |
| EMA average | 0.255 | **0.371** | **0.322** |

## F    IMPLEMENTATION DETAILS

The detailed algorithmic procedure for HiGS is provided in Algorithm 1, with pseudocode given in Algorithms 2 and 3. Since HiGS reuses past predictions without requiring additional forward passes, its computational cost is equivalent to standard CFG. Other operations, such as DCT, add negligible overhead. Thus, HiGS improves quality without increasing the overall sampling cost or memory as can be seen in the code and our experiments.

We always scale the time step $t$ such that $t \in [0, 1]$. For the frequency filter, we found that $\lambda = 50$ and $R_c \approx 0.05$ work well across all models. Similarly, setting $t_{\min} \in [0.3, 0.5]$ and $t_{\max} \in [0.9, 1.0]$ lead

Table 10: Guidance parameters used for Table 1.

| Model | # Steps | $w_{\text{CFG}}$ | $w_{\text{HiGS}}$ | $\eta$ | $t_{\min}$ | $t_{\max}$ | $\alpha$ | $R_c$ |
|---|---|---|---|---|---|---|---|---|
| SiT-XL + REPA | 30 | 1.5 | 1 | 1 | 0.3 | 1 | 0.75 | 0.05 |
| DiT-XL/2 | 15 | 1.25 | 2 | 0 | 0.3 | 1 | 0.75 | 0.05 |
| Stable Diffusion XL | 20 | 2.5 | 1.75 | 0 | 0.4 | 1 | 0.75 | 0.05 |
| Stable Diffusion 3 | 20 | 2.5 | 1.75 | 0 | 0.4 | 1 | 0.75 | 0.05 |

Table 11: Guidance parameters used for Table 2.

| Model | # Steps | $w_{\text{CFG}}$ | $w_{\text{HiGS}}$ | $\eta$ | $t_{\min}$ | $t_{\max}$ | $\alpha$ | $R_c$ |
|---|---|---|---|---|---|---|---|---|
| Stable Diffusion XL | 20 | 2.5 | 1.75 | 1 | 0.4 | 1 | 0.5 | 0.05 |
| Stable Diffusion 3 | 20 | 2.5 | 1.75 | 1 | 0.4 | 1 | 0.5 | 0.05 |
| Stable Diffusion 3.5 | 20 | 2.5 | 1.75 | 1 | 0.4 | 1 | 0.5 | 0.05 |

Table 12: Guidance parameters used for Table 3.

| Model | # Steps | $w_{\text{CFG}}$ | $w_{\text{HiGS}}$ | $\eta$ | $t_{\min}$ | $t_{\max}$ | $\alpha$ | $R_c$ |
|---|---|---|---|---|---|---|---|---|
| SiT-XL + REPA (Unguided) | 40 | 1 | 1 | 1 | 0.4 | 1 | 0.75 | 0.05 |
| SiT-XL + REPA (with CFG) | 40 | 1.8 | 1 | 1 | 0.35 | 1 | 0.75 | 0.05 |
| SiT-XL + REPA-E (Unguided) | 30 | 1 | 1.25 | 1 | 0.35 | 1 | 0.75 | 0.05 |
| SiT-XL + REPA-E (with CFG) | 40 | 2.5 | 0.75 | 0 | 0.3 | 1 | 0.75 | 0.05 |

to consistently strong results in our tests. For EMA values, we observed that $\alpha = 0.5$ or $\alpha = 0.75$ worked well across all experiments. We used all past predictions as the history length $W$, which allows us to compute the EMA average on the fly without buffering all previous predictions in memory. The hyperparameters used for each experiment are given in Tables 10 to 12.

For quantitative evaluation, FID scores for class-conditional models in Table 1 are reported using 10,000 generated samples and the full ImageNet training set. The ImageNet results in Table 3 are based on 50,000 generated samples to ensure fair comparison with prior work. For text-to-image models, we used the entire validation set of the COCO 2017 dataset (Lin et al., 2014) as ground truth text-image pairs. We followed the ADM evaluation framework (Dhariwal & Nichol, 2021) for computing FID, IS, Precision, and Recall to maintain consistency across all evaluations.

## G  ADDITIONAL VISUAL EXAMPLES

We provide additional visual examples in this section to demonstrate the effectiveness of HiGS in enhancing the quality of various models across a wide range of guidance scales and sampling setups. Figures 16 to 19 show further text-to-image generation results using Stable Diffusion models (Podell et al., 2023; Esser et al., 2024). In addition, Figure 20 presents visual results for applying HiGS to Flux (Labs, 2024). Finally, Figure 21 provides class-conditional generation with SiT-XL + REPA (Yu et al., 2025a). In all cases, HiGS consistently improves quality over the baseline.

---

**Algorithm 1** Sampling with HiGS

---

**Require:** Diffusion model $D_{\boldsymbol{\theta}}$, input condition $\boldsymbol{y}$
**Require:** CFG scale $w_{\text{CFG}}$, HiGS schedule $w_{\text{HiGS}}(t)$, history length $W$
**Require:** EMA parameter $\alpha$, projection weight $\eta$, DCT parameters $(R_c, \lambda)$
 1: Initialize latent $\boldsymbol{z}_T \sim \mathcal{N}(\boldsymbol{0}, \boldsymbol{I})$, history buffer $\mathcal{H} \leftarrow \emptyset$
 2: **for** $t_k \in \{t_0, t_1, \ldots, t_T\}$ **do**
 3:     Compute conditional and unconditional predictions: $D_c(\boldsymbol{z}_{t_k}), D_u(\boldsymbol{z}_{t_k})$
 4:     Apply CFG: $D_{\text{CFG}}(\boldsymbol{z}_{t_k}) = w_{\text{CFG}} D_c(\boldsymbol{z}_{t_k}) - (w_{\text{CFG}} - 1) D_u(\boldsymbol{z}_{t_k})$
 5:     Compute the history signal $g(\mathcal{H}_k) = \sum_{i \in I_k} \alpha (1 - \alpha)^{t-1-i} D_{\text{CFG}}(\boldsymbol{z}_{t_i})$
 6:     Form the guidance direction: $\Delta D_{t_k} = D_{\text{CFG}}(\boldsymbol{z}_{t_k}) - g(\mathcal{H}_k)$
 7:     **if** projection enabled (i.e., $\eta < 1$) **then**
 8:         Project $\Delta D_{t_k}$ into orthogonal and parallel components w.r.t. $D_{\text{CFG}}(\boldsymbol{z}_{t_k})$
 9:         $\Delta D_{t_k} \leftarrow \Delta D_{t_k}^{\perp} + \eta \Delta D_{t_k}^{\parallel}$
10:     **end if**
11:     Apply DCT-based high-pass filter: $\Delta D_{t_k} \leftarrow \text{iDCT}(H(R) \cdot \text{DCT}(\Delta D_{t_k}))$
12:     Apply HiGS: $D_{\text{HiGS}}(\boldsymbol{z}_{t_k}) = D_{\text{CFG}}(\boldsymbol{z}_{t_k}) + w_{\text{HiGS}}(t_k) \Delta D_{t_k}$
13:     Update history: $\mathcal{H}_{k+1} \leftarrow \mathcal{H}_k \cup \{D_{\text{CFG}}(\boldsymbol{z}_{t_k})\}$, truncate oldest if $|\mathcal{H}_{k+1}| > W$
14:     Apply one sampling step: $\boldsymbol{z}_{t-1} = \text{SAMPLINGSTEP}(D_{\text{HiGS}}(\boldsymbol{z}_t), \boldsymbol{z}_t, t)$
15: **end for**
16: **return** $\boldsymbol{z}_0$

---

**Algorithm 2** Utility functions used in the implementation of HiGS.

```python
import torch
import torch_dct as dct

def project(
    v0: torch.Tensor,  # [B, C, H, W]
    v1: torch.Tensor,  # [B, C, H, W]
):
    dtype = v0.dtype
    v0, v1 = v0.double(), v1.double()
    v1 = torch.nn.functional.normalize(v1, dim=[-1, -2, -3])
    v0_parallel = (v0 * v1).sum(dim=[-1, -2, -3], keepdim=True) * v1
    v0_orthogonal = v0 - v0_parallel
    return v0_parallel.to(dtype), v0_orthogonal.to(dtype)

def square_root_schedule(t, start=0, end=1):
    if t > end or t <= start:
        return 0.0
    return ((t - start) / (end - start)) ** 0.5

def dct2(x):
    return dct.dct_2d(x, norm="ortho")

def idct2(x):
    return dct.idct_2d(x, norm="ortho")

def apply_high_freq_dct_mask(diff, threshold=0.05, sharpness=50):
    B, C, H, W = diff.shape
    device = diff.device
    X = dct2(diff)
    u = torch.arange(H, device=device).view(H, 1) / H
    v = torch.arange(W, device=device).view(1, W) / W
    d = torch.sqrt(u**2 + v**2)  # normalized distance from top-left (DC)
    mask = torch.sigmoid((d - threshold) * sharpness)
    X_filtered = X * mask  # broadcast over (B, C)
    diff_filtered = idct2(X_filtered).to(diff.dtype)
    return diff_filtered

class HistoryBuffer:
    def __init__(self, ema_alpha=0.75):
        self.ema = None
        self.ema_alpha = ema_alpha

    def add(self, current_pred):
        if self.ema is None:
            self.ema = torch.zeros_like(current_pred)
        self.ema = self.ema_alpha * current_pred + (1 - self.ema_alpha) * self.ema
```

**Algorithm 3** PyTorch implementation of HiGS.

```python
class HiGSGuidance:
    def __init__(
        self,
        w_higs,
        t_min=0.4,
        t_max=1.0,
        eta=1.0,
        ema_alpha=0.75,
        dct_threshold=0.05,
    ):
        self.history = HistoryBuffer(ema_alpha=ema_alpha)
        self.weight = w_higs
        self.min_t = t_min
        self.max_t = t_max
        self.parallel_weight = eta
        self.dct_threshold = dct_threshold

    def step(self, current_pred, timestep=None):
        """
        Compute the HiGS guidance step.
        """
        # current_pred can be either CFG-guided or conditional predictions
        if self.history.ema is None:
            self.history.add(current_pred)
            return torch.zeros_like(current_pred)

        diff = current_pred - self.history.ema

        # compute the projection of the difference
        diff_par, diff_orth = project(diff, current_pred)
        diff = diff_orth + diff_par * self.parallel_weight

        # Compute the scaling factor based on the current timestep
        gamma = square_root_schedule(timestep, self.min_t, self.max_t)
        scale = self.weight * gamma

        # Update the history with the current prediction
        self.history.add(current_pred)

        # Apply the high-frequency DCT mask to the difference
        if self.dct_threshold >= 0:
            diff = apply_high_freq_dct_mask(diff, threshold=self.dct_threshold)

        # Return the scaled difference
        return scale * diff
```

| CFG | + HiGS (Ours) | CFG | + HiGS (Ours) |

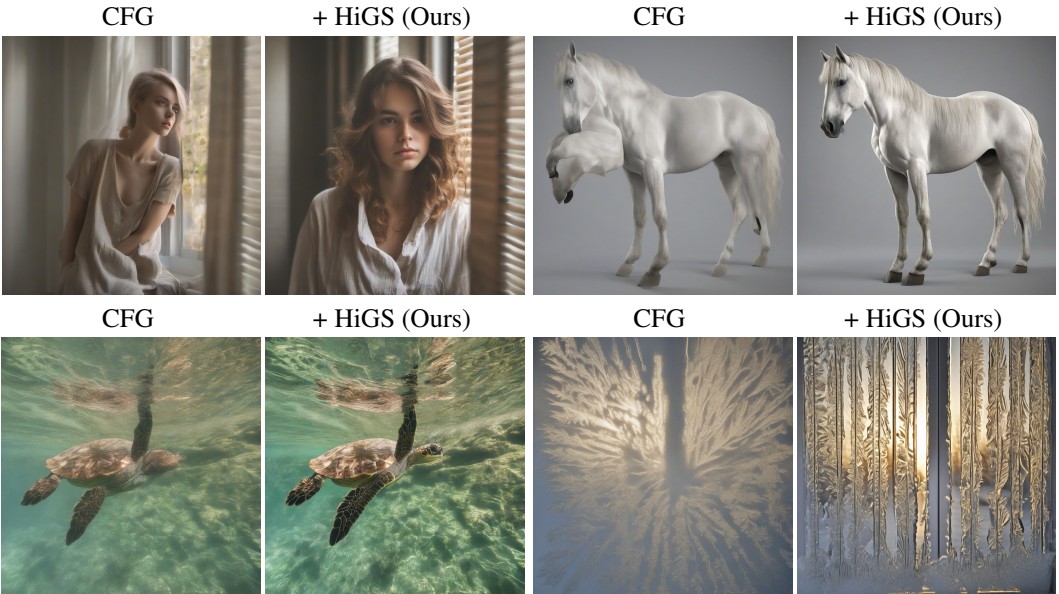

Figure 16: Generated samples using Stable Diffusion XL with 20 steps and $w_{\mathrm{CFG}} = 3$.

| CFG | + HiGS (Ours) | CFG | + HiGS (Ours) |

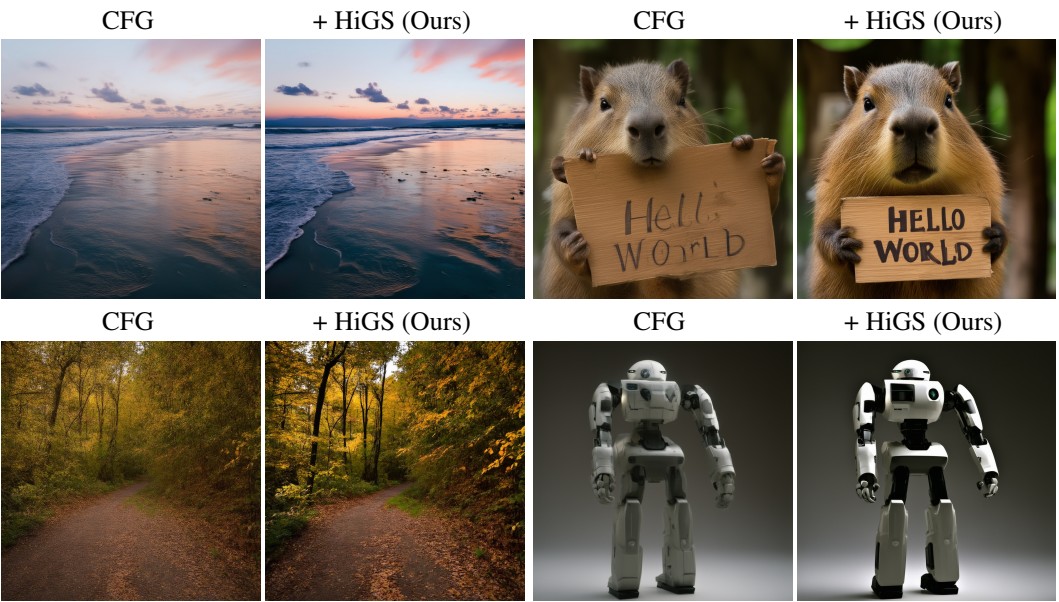

Figure 17: Generated samples using Stable Diffusion 3 with 16 steps and $w_{\mathrm{CFG}} = 2$.

CFG      + HiGS (Ours)      CFG      + HiGS (Ours)

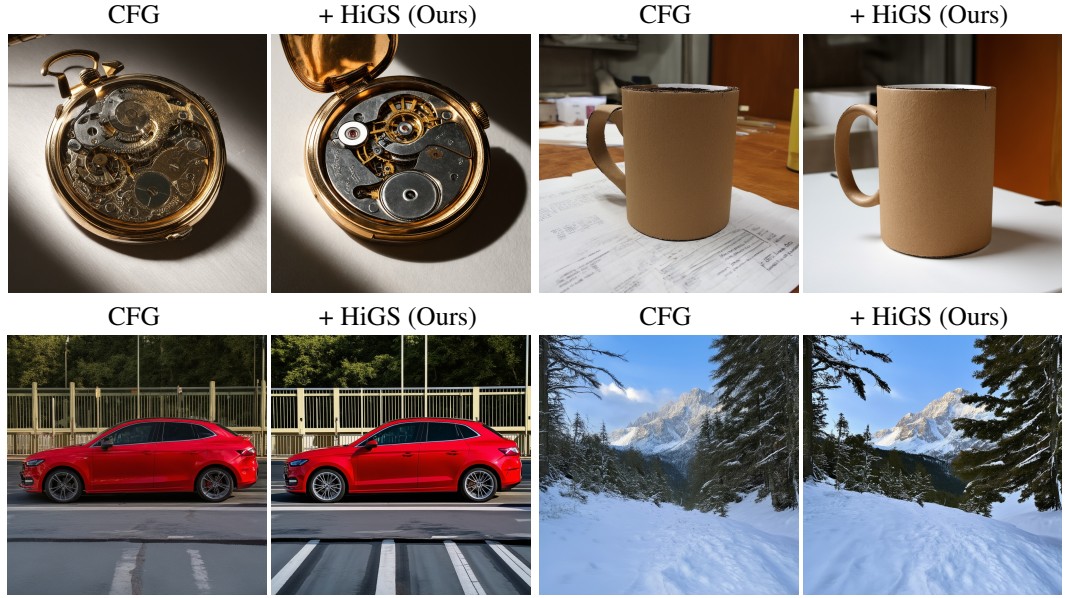

CFG      + HiGS (Ours)      CFG      + HiGS (Ours)

Figure 18: Generated samples using Stable Diffusion 3 with 28 steps and $w_{\text{CFG}} = 4.5$.

CFG      + HiGS (Ours)      CFG      + HiGS (Ours)

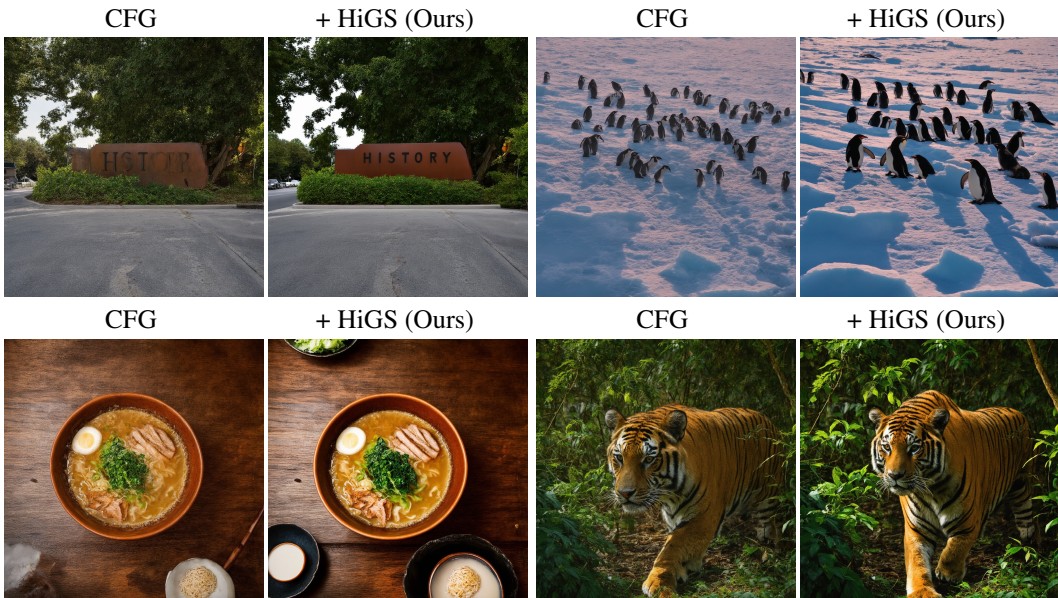

CFG      + HiGS (Ours)      CFG      + HiGS (Ours)

Figure 19: Generated samples using Stable Diffusion 3 with 28 steps and $w_{\text{CFG}} = 2$.

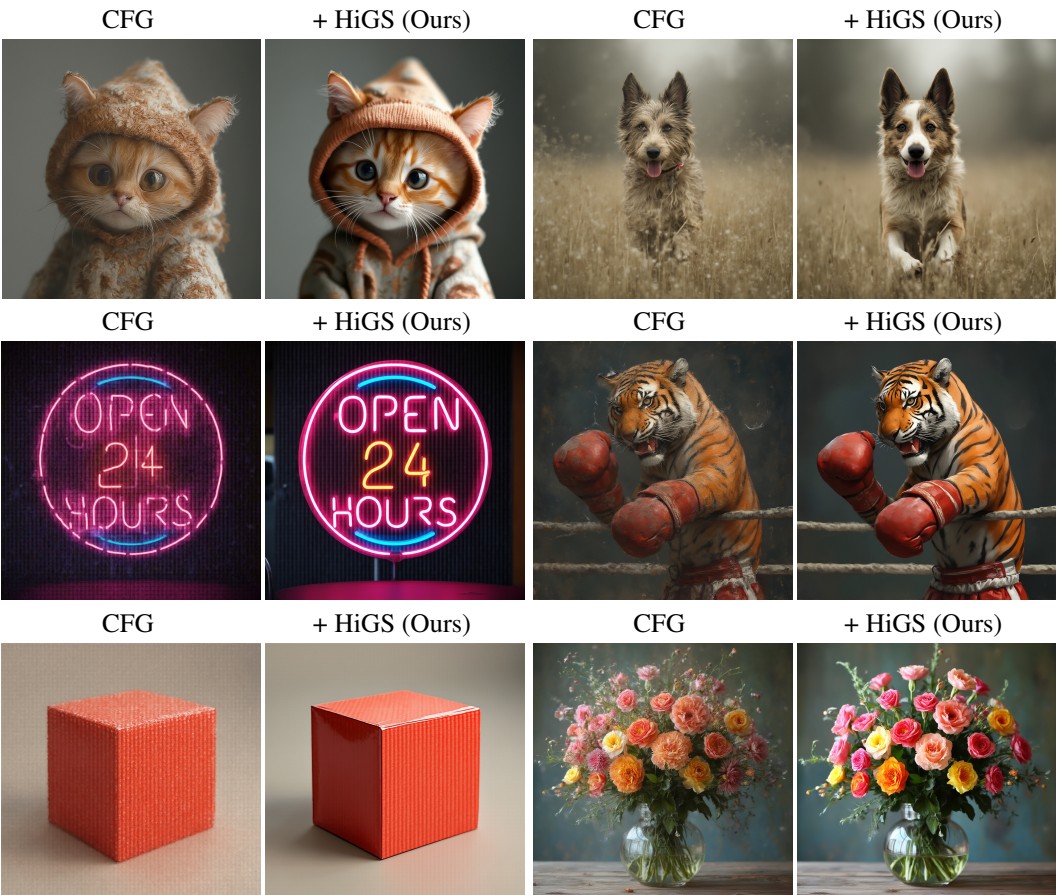

Figure 20: Generated samples using Flux with 15 steps and $w_{\text{CFG}} = 1.25$.

CFG

+ HiGS (Ours)

CFG

+ HiGS (Ours)

CFG

+ HiGS (Ours)

CFG

+ HiGS (Ours)

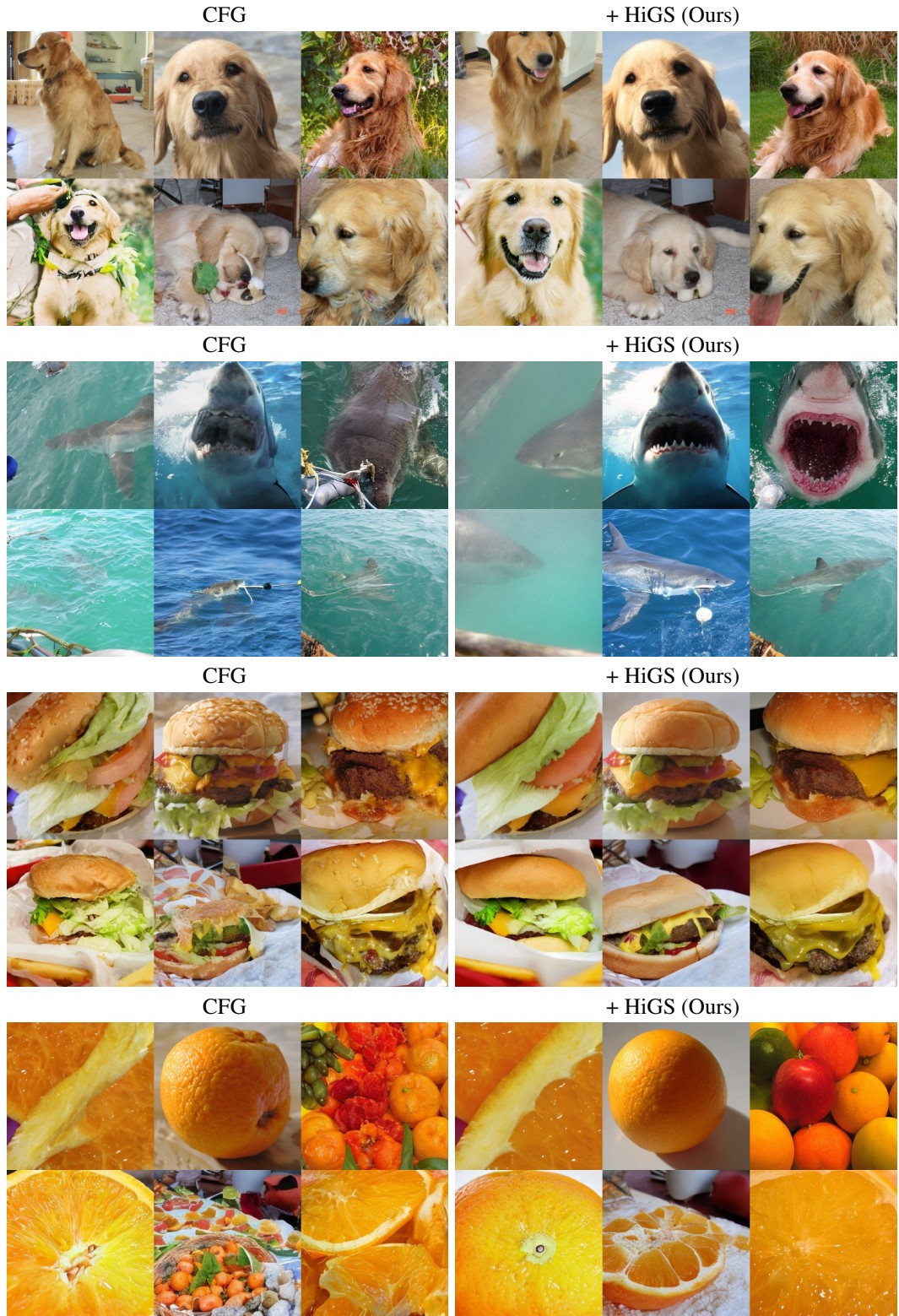

Figure 21: Class-conditional generation with SiT-XL + REPA using 30 steps and $w_{\text{CFG}} = 1.8$.

