# OpenReview forum: "HiGS: History-Guided Sampling for Plug-and-Play Enhancement of Diffusion Models"
_ICLR.cc/2026/Conference — ICLR 2026 Poster_

### Official Review · Reviewer_2itd · 2025-10-20

**Soundness:** 4
**Presentation:** 3
**Contribution:** 3
**Rating:** 8
**Confidence:** 4

**Summary:**

The author introduces a novel momentum-based sampling technique called History-Guided Sampling (HiGS), which improves the quality and efficiency of diffusion sampling by incorporating recent model predictions into each inference step. This approach yields more realistic outputs with finer details. Experiments across various diffusion models and datasets demonstrate promising improvements.

**Strengths:**

* The motivation is compelling: the authors propose that the Euler sampler used in diffusion models can be interpreted as performing stochastic gradient descent (SGD) on a time-varying energy function. Based on this perspective, they argue that the gradient estimate can be enhanced to improve both the efficiency and quality of the sampling process, incorporating a momentum-based term inspired by STORM.

* HiGS is conceptually straightforward and can be easily integrated into pre-trained diffusion models. Moreover, it introduces minimal additional computational overhead.

* The experiments are comprehensive in terms of datasets and model types, and the performance is persuasive.

**Weaknesses:**

* In Section 4.2, the methodology is presented in a relatively loose and unstructured manner. It would be beneficial to include clearer explanations of the design choices and a more thorough ablation study to identify which components contribute most to performance. Additionally, clarifying the rationale behind how these techniques are organized and integrated would strengthen the overall presentation.

* The tuning required for the combination of techniques in the HiGS design may pose a practical burden for users, potentially limiting its accessibility and ease of adoption.

**Questions:**

* What motivated the choice of Equation 5 as the formulation for HiGS over other possible alternatives?

* Could you elaborate on why "the benefits of HiGS were most evident during the early and middle sampling steps," as stated in line 226? What underlying factors contribute to this behavior?

---

> ### Author Response · Authors · 2025-11-18
>
> We thank the reviewer for the strong and positive evaluation of our submission and for acknowledging its key strengths, including the motivation, its universal nature, and its extensive evaluation. We have provided our responses to the comments below, and we are open to further discussion.
>
> ###  **Method presentation in Section 4.2**
> We thank the reviewer for the suggestion. Due to page-limit constraints, we were unable to include the full set of ablations in the main paper and therefore focused Section 4.2 on describing the components of our method. However, we have provided extensive ablation studies and detailed analysis in Section E to isolate the contribution of each component and to clarify the underlying design choices. Since the camera-ready version allows for 10 pages, we would be happy to use the additional space to further justify our design choices in the main text.
>
> ###  **Tuning requirements for HiGS**
> We would like to emphasize that, as shown in Section E, HiGS is not highly sensitive to its underlying parameters, and multiple configurations yield good performance. We have also provided reasonable baseline values for each parameter in Section F and in Algorithm 2, and we believe these values serve as strong starting points when adapting HiGS to new models.
>
> ### **Motivation behind the choice of Equation 5**
> This is motivated by how EMA-momentum is used in stochastic optimization (such as Adam) to improve performance. We have also tested other formulations in Section E and  showed that HiGS is flexible w.r.t. this design choice.
>
> ### **Effect of HiGS in various time-steps**
> We thank the reviewer for this question and would like to provide further clarification. Since the structure and content of the image are primarily formed during the early to mid sampling steps of diffusion models, we find that these are also the stages where HiGS is most effective at guiding the trajectory toward better generations [1, 2]. In the later sampling steps (i.e., at low noise levels), the image is already largely determined in terms of details and structure, making it difficult to correct mistakes at that point. This behavior is consistent with CFG, which also has a strong influence during early and mid sampling steps but has minimal impact in the final stages [1]. Based on these observations, we place stronger emphasis on applying HiGS during the early to mid phases of sampling.
>
> [1] Kynkäänniemi T, Aittala M, Karras T, Laine S, Aila T, Lehtinen J. Applying guidance in a limited interval improves sample and distribution quality in diffusion models. Advances in Neural Information Processing Systems. 2024 Dec 16;37:122458-83.
>
> [2] Nath V, Lau E, Gunjal A, Sharma M, Baharte N, Hendryx S. Adaptive Guidance Accelerates Reinforcement Learning of Reasoning Models. arXiv preprint arXiv:2506.13923. 2025 Jun 16.

---

> > ### Comment · Reviewer_2itd · 2025-11-19
> >
> > Thank you for the responses.

---

### Official Review · Reviewer_etRw · 2025-10-28

**Soundness:** 1
**Presentation:** 2
**Contribution:** 1
**Rating:** 4
**Confidence:** 5

**Summary:**

This paper introduces History-Guided Sampling (HiGS), a novel, momentum-based sampling technique designed to enhance the image quality and sampling efficiency of existing diffusion models. The core insight is that integrating the history of recent model predictions into the current inference step can steer the process toward more realistic outputs with improved fine details, particularly when using a low number of neural function evaluations (NFEs) or lower classifier-free guidance (CFG) scales. The method leverages the difference between the current noise prediction and a weighted average of past predictions. Crucially, HiGS operates as a plug-and-play enhancement, requiring no additional training or fine-tuning of the base diffusion model, and introduces negligible computational overhead. The authors demonstrate HiGS's effectiveness across various models (including Stable Diffusion 3 and SiT) and achieve a new state-of-the-art FID of 1.61 using a pretrained SiT model, showcasing both strong efficiency and quality gains.

**Strengths:**

1. High Utility and Plug-and-Play Nature: The primary strength is that HiGS is a highly practical and post-hoc sampling enhancement. It requires no modifications to the pre-trained diffusion model or additional training, making it universally applicable and easy to integrate into existing workflows.

2. Exceptional Efficiency and Quality: The method boosts image quality even under highly constrained inference budgets (low NFEs). By leveraging a simple weighted average of history predictions, the approach introduces practically no additional computational cost.

3. Effectiveness: HiGS is shown to be effective across diverse models and architectures (SD3, SiT, SDXL-Flash), indicating that the concept of "history-guided" momentum is a fundamental enhancement to the general sampling process, rather than a solution tailored to one specific model.

**Weaknesses:**

1. Lack of Novelty and Differentiation from Prior Work (UCGM [1]): The approach bears significant conceptual similarity to prior work, specifically UCGM (see [1]). The paper needs to provide a rigorous comparison (both theoretical and empirical) to clearly articulate how HiGS fundamentally differs from and improves upon UCGM's methodology, particularly in its formulation of the history/momentum term.

2. Inconsistency in Theoretical Formulation and Experiments: The theoretical foundation is established using the EDM (Elucidated Diffusion Models) formulation, yet the experimental evaluation is conducted on models trained using different frameworks (DDPM for DiT/SDXL, Flow Matching for SiT/SD3). This mismatch between the theoretical background and the experimental setup must be reconciled. The authors should either generalize the theory to cover the different diffusion formulations used or provide an EDM-based experimental validation.

3. Weak or Misleading Theoretical Justification (SGD/Energy): The introduction of concepts related to Stochastic Gradient Descent (SGD) and energy functions to justify the momentum term (e.g., logic around lines 186-187) appears tenuous and possibly forced ("shoehorned"). The authors must critically re-evaluate and simplify the theoretical derivation to establish a clear, direct, and convincing link between the proposed history-guided mechanism and the underlying physics/optimization, or remove the irrelevant theoretical connections.

4. Insufficient Interaction Analysis in Ablation Study: The ablation studies (e.g., Tables 7, 8, and 9) are presented linearly and fail to adequately analyze the interaction between the various design components. It is unclear how designs are built upon each other (e.g., whether Table 8 is based on Table 7, or how $g(H_k)$ in Table 9 integrates with prior designs). The authors must clarify the baseline for each subsequent ablation and provide more comprehensive, possibly combinatorial, results to demonstrate the synergistic or interdependent effects of their design choices.

5. Theoretical Justification of the Momentum Term: While the mechanism is clearly defined, the paper should provide a deeper theoretical analysis or intuition explaining why the divergence from a historical average (the "difference between the current prediction and a weighted average of past predictions") systematically leads to higher quality, more detailed results. Is it effectively smoothing the variance of the noise prediction, or is it introducing a form of implicit guidance toward the average manifold learned during training?

6. Hyperparameter Sensitivity Analysis: As a momentum-based technique, HiGS inherently introduces a set of new parameters (e.g., the weighting factor for the history, the depth or length of the history used). The paper needs to include a thorough sensitivity analysis demonstrating how robust the results are to changes in these hyperparameters. Are the reported gains dependent on a finely tuned set of parameters?

7. Potential for Temporal Artifacts/Bias: I would like to see discussion or analysis on whether incorporating prediction history could inadvertently introduce minor temporal correlations or specific texture biases into the generated images, even if the overall FID improves. A few qualitative examples showing cases where HiGS fails or introduces unintended artifacts (if any exist) would strengthen the paper.


[1] Unified Continuous Generative Models. In page 7 of  https://arxiv.org/pdf/2505.07447, it proposes **Extrapolating the estimation**, which shares the same idea as this paper.

**Questions:**

See weaknesses.

---

> ### Author Response · Authors · 2025-11-18
>
> We thank the reviewer for providing thoughtful feedback on our paper and for recognizing its strengths, including its plug-and-play nature, effectiveness, and exceptional efficiency and quality. We have provided our responses to the reviewer’s comments below and welcome further discussion.
>
> ### **Relation to UCGM**
> We appreciate the reviewer’s pointing out this promising and concurrent line of work. While both UCGM and HiGS share the idea of leveraging history to improve sampling, there are several differences in design and scope that distinguish the two methods:
>
> - **Frequency filtering and projection:** We found that both frequency filtering and projection are essential for the performance of HiGS and for avoiding saturation and color artifacts (as shown in Section E). UCGM does not address these key components or issues in the main paper.
> - **Use of history:** UCGM only uses the previous prediction, whereas our method uses an EMA of past predictions. We found that using an EMA improves sampling compared to using only the previous prediction (HPSv2: 0.261 vs. 0.251; ImageReward: 0.40 vs. 0.33).
> - **Weight scheduling:** UCGM uses a fixed, low weight (e.g., 0.5), while we found that using higher weights together with scheduling is essential for improving quality in HiGS (HPSv2: 0.261 vs. 0.245).
> - **Scope of evaluation:** UCGM is evaluated only on ImageNet generation, whereas we provide extensive evaluations on more challenging text-to-image tasks. We also show the synergy between HiGS and classifier-free guidance and demonstrate compatibility with multiple diffusion samplers, all of which are absent in UCGM.
> - **Theoretical analysis:** Our theoretical analysis supports time-varying step sizes (which are more realistic for diffusion models), while UCGM provides analysis only for uniform time steps.
>
> Overall, we view UCGM as complementary and concurrent work to ours, and we hope that highlighting these distinctions clarifies the unique aspects contributed by HiGS.
>
> ### **Inconsistency in theoretical formulation**
> We appreciate the reviewer’s attention to this point. Using the EDM formulation in our theoretical section is not a limitation, as any diffusion model can be expressed within this unified framework with appropriate definitions (e.g., see Table 1 in [1] and Section 2 and 3 in [2]). We therefore believe this formulation provides clarity rather than inconsistency.
>
> [1] Elucidating the Design Space of Diffusion-Based Generative Models
>
> [2] Scaling rectified flow transformers for high-resolution image synthesis
>
> ### **Weak or misleading theoretical justification (SGD/Energy)**
> We respectfully disagree with the reviewer on this point and would like to emphasize that the connection we draw in Section 4.1 follows directly from the definitions of the score and energy functions. The resulting insights have also proven useful in practice, particularly in motivating the use of history as a guidance signal that improves generation quality. We would be happy to further expand this explanation in the final version of the paper.
>
> ### **Insufficient interaction analysis in ablation study**
> Our ablations show that HiGS is not highly sensitive to its parameters (e.g., Figure 14 and 15, Table 8 and 9), and several good configurations lead to improved performance. We also provide reasonable default values in Section F and Algorithm 2, which we have found to be effective across almost all models. Conducting a full combinatorial ablation via grid search would require significant computational resources that fall outside the scope of the current submission.
>
> ### **Theoretical justification of the momentum term**
> Intuitively, the difference between the current prediction and the history highlights the high-frequency components added at each sampling step, offering strong corrective directions. Our theoretical analysis in Section B also shows that incorporating such guidance terms can reduce the approximation error of the ODE solver.
>
> ### **Hyperparameter sensitivity**
> Section F includes an in-depth analysis of how each parameter in HiGS affects the final generation quality. We note that HiGS is not highly sensitive to its parameters (e.g., Figure 14 and 15, Table 8 and 9), and several good configurations lead to improved performance. In addition, we have provided reasonable default values for each parameter in Section F and Algorithm 2.
>
> ### **Potential for temporal artifacts**
> In our experiments, we have not observed noticeable temporal artifacts when applying HiGS, which is also reflected in the qualitative examples included in the submission. As with CFG, very large weights may introduce noisy behavior, and we can include this discussion in the final version of the paper for completeness.

---

> ### Comment · Reviewer_etRw · 2025-11-27
>
> Thank you for your detailed responses. There are a few more questions I have after reading your rebuttal:
>
>
> 1. Theoretical-Experimental Mismatch in EDM Formulation
>
> Regarding the EDM formulation: Reference [1] establishes that the EDM formulation corresponds to variance-exploding SDE (VE-SDE) [2], whereas DDPM and flow matching are grounded in variance-preserving SDE (VP-SDE). Intrinsically, VE-SDE and VP-SDE exhibit distinct sampling characteristics [2], which indicates that **theoretical results derived for VP-SDE cannot be directly transferred to VE-SDE settings**. Consequently, there appears to be a fundamental mismatch between the theoretical framework presented in the paper and the experimental validation. **I strongly recommend conducting additional experiments using an EDM-based setup to properly validate the theoretical contributions and ensure consistency between theory and practice.**
>
> 2. Anomalous Results in Table 1
>
> Table 1 presents results that appear inconsistent with established findings in the literature. Prior work [3,4] has demonstrated that SiT exhibits faster convergence than DiT, with REPA providing further acceleration. However, **Table 1 shows that SiT-XL+REPA achieves worse FID performance than DiT-XL/2, which contradicts these well-established results**. To clarify this discrepancy, please provide:
>
> - The specific checkpoints used for both SiT-REPA and DiT (whether official checkpoints were utilized)
> - Complete configurations and hyperparameters
> - A detailed explanation of the unexpected performance gap
>
> Without this information, it is difficult to assess whether these results stem from implementation issues, suboptimal hyperparameters, or represent genuine findings that challenge existing understanding.
>
> 3. Inconsistency in Guidance Parameter Reporting
>
> I have identified an internal contradiction in the manuscript regarding guidance parameters. Line 394 states: "sampling with and without HiGS is performed using the same NFE and CFG scale." However, Table 10 reveals that the guidance parameters for CFG and HiGS differ substantially. For instance, SiT-XL and SiT-XL+REPA use $w_\text{CFG}=1.5$ versus $w_\text{HiGS}=2.0$. This discrepancy contradicts the claim of using identical guidance scales and raises concerns about the fairness of the comparison. Please either revise the statement in line 394 or provide justification for using different guidance parameters while claiming equivalence.
>
> 4. Performance Saturation and Upper Bound of HiGS
>
> Table 3 demonstrates that HiGS achieves comparable performance to 250-step CFG using only 40 sampling steps. However, it remains unclear whether HiGS performance saturates at this point or continues to improve with additional steps. Specifically:
>
> - Does FID consistently decrease as the number of sampling steps increases beyond 40?
> - What is the performance ceiling of HiGS when computational budget is not a constraint?
> - How does the FID-NFE relationship of HiGS compare to CFG across a broader range of sampling steps?
>
>
>
> [1] Simplifying, Stabilizing and Scaling Continuous-Time Consistency Models.
>
> [2] Score-Based Generative Modeling through Stochastic Differential Equations.
>
> [3] SiT: Exploring Flow and Diffusion-based Generative Models with Scalable Interpolant Transformers
>
> [4] Representation alignment for generation: Training diffusion transformers is easier than you think.

---

> > ### Author Response · Authors · 2025-11-27
> >
> > We thank the reviewer for their engagement. Please find our responses to the questions below, and we remain open to further discussion.
> >
> > ### **Theoretical-experimental mismatch in EDM formulation**
> > We respectfully point out that the ODE formulation $dz/dt = -u(z,t)$ (see Appendix B) used in our work covers nearly all modern diffusion frameworks, including EDM, DDPM, and flow-matching models. Despite differences in training objectives, current formulations (such as VP, VE, DDPM, and DDIM) can all be interpreted as special cases of this unified ODE, as summarized in Table 1 of [1]. Accordingly, all our experiments were carried out under this same generic ODE formulation, and we therefore do not believe there is a mismatch between our theoretical analysis and practical results.
> >
> > [1] Karras T, Aittala M, Aila T, Laine S. Elucidating the design space of diffusion-based generative models. Advances in neural information processing systems. 2022 Dec 6;35:26565-77.
> >
> > ### **Anomalous results in Table 1**
> >
> > We confirm that all models were evaluated using their official checkpoints and hyperparameters. The *only* deviation from the official codebases is the number of sampling steps. Following the reviewer’s question, we observed the following implementation detail that explains the behavior in Table 1:
> >
> > - SiT models use “guidance interval’’ by default, meaning CFG is applied only to a subset of sampling steps.
> > - DiT applies CFG at every step.
> >
> > While guidance interval improves quality at high NFE, it seems that it can adversely impact the quality of the baseline at low NFE, which is the regime evaluated in Table 1. However, note that SiT significantly outperforms DiT after applying HiGS, which aligns with the intuition that flow-matching models have higher potential at lower sampling budgets.
> >
> > ### **Inconsistency in guidance parameter reporting**
> > We wish to correct the misunderstanding regarding our parameter reporting. HiGS operates as an additive correction on top of the CFG output (see Equation 9); therefore, the parameter $w_\text{HiGS}$ represents the “history guidance weight”, which is a distinct and separate hyperparameter introduced by our method (see Algorithm 2). Accordingly, Sampling with and without HiGS utilizes the “identical CFG scale” ($w_\text{CFG}$) in our paper to ensure a controlled experimental setup. We do not alter the base $w_\text{CFG}$ when comparing baselines to HiGS. We will revise the manuscript to prevent future confusion in this regard.
> >
> > ### **Performance saturation and upper bound of HiGS**
> > As shown in Figure 5, HiGS maintains a clear margin over the baseline at the default sampling steps for Stable Diffusion 3. We therefore believe that HiGS offers notable benefits even at higher sampling budgets. This is further supported by Table 3, where HiGS with only 30 steps already outperforms the original sampler with 250 steps (FID 1.83 vs. 1.61). That being said, we expect the performance of HiGS to gradually approach a limiting point as the number of sampling steps becomes very large, similar to other diffusion samplers. Given the time constraint of the rebuttal, we are happy to explore this discussion further in the final version of our paper.

---

### Official Review · Reviewer_zxKC · 2025-10-30

**Soundness:** 3
**Presentation:** 3
**Contribution:** 2
**Rating:** 6
**Confidence:** 4

**Summary:**

The paper proposes History-Guided Sampling, a training-free, momentum-style modification to diffusion sampling. It injects a correction term at each step using an EMA of past denoiser predictions. The update is scheduled over time, optionally projected to suppress the parallel component to avoid oversaturation, and high-pass filtered in the DCT domain to prevent color drift. The authors motivate HiGS by viewing Euler sampling as SGD on a time-varying energy and borrowing a STORM-like momentum term, as well as offering an error analysis showing improved local truncation error and thus better global error order for the modified update. The experiments show some improvements of HiGS compared with Stable Diffusion variants, DiT/SiT with low NFE and low CFG. It is notable that the unguided HiGS reports FID 1.61 in 30 steps on ImageNet-256.

**Strengths:**

1.	The paper introduces a training-free “history” term during sampling (EMA over past predictions) and combines it with two stabilizers—orthogonal projection and DCT high-pass filtering. The combination is simple in this plug-and-play form, and well-motivated by both optimization (STORM-style momentum) and numerical analysis views.
2.	The paper gives two theoretical lenses: (1) Euler sampling ≈ SGD on a time-varying energy, motivating history/momentum; (2) a truncation-error analysis showing that a properly weighted history term can improve the local error from O(h^2) to O(h^3) and thus the global error order.
3.	The experiments show consistent gains across models including SDXL/SD3/SD3.5, DiT/SiT, tasks, and multi samplers. Quantitative tables show large preference-metric jumps and lower FID/IS steps to match baselines; ImageNet results include a new unguided FID 1.61 in 30 steps on SiT-XL+REPA-E

**Weaknesses:**

1.	There may be a gap between the clean theory and the implemented recipe. The error-order analysis assumes an ideal history weight $w_k$ and doesn't account for EMA history, time-windowing, orthogonal projection, or DCT filtering used in practice; the proof shows existence of a good $w_k$, not that the EMA-driven schedule approximates it.
2.	Text-to-image improvements are primarily via HPSv2/ImageReward/Win-rate; CLIP stays flat in Table 5. There is no formal human evaluation or statistical significance tests across prompts or seeds.

**Questions:**

1.	The error analysis (App. B) shows an existence result for a weight $w_k=\frac{h_k}{2 h_{k-1}}$ that upgrades local error to $O(h_k^3)$. In practice the paper use an EMA history, a scheduled $w_{\text {HiGS }}(t)$, projection, and DCT filtering. Can the author quantify how closely the implemented EMA + schedule approximates the ideal $w_k$ ? A small toy ODE or linearized score model comparison would help.
2.	Section 4.1 frames Euler as SGD on a time-varying energy and connects the history term to STORM style gradient differences. Can the authors provide an empirical plot of $\left\|\nabla E_t\left(z_t\right)-\nabla E_{t-1}\left(z_{t-1}\right)\right\|$ vs. the EMA residual magnitude during sampling to validate the link?
3.	The theorem assumes smoothness and bounded derivatives. Are there empirical signs (e.g., at extreme CFG scales) that these assumptions break and the history term can destabilize?

---

> ### Author Response · Authors · 2025-11-18
>
> We thank the reviewer for providing feedback on our submission and for acknowledging our contributions, including the extensive evaluation, the well-motivated methodology, and the theoretical explanations. We respond to each of the reviewer’s comments below and are open to further discussion.
>
> ### **Gap between theory and the implementation**
> We acknowledge that a few assumptions are necessary in order to provide a meaningful analysis of the underlying mechanisms of HiGS. In practice, we found that additional components of HiGS further improve performance, since real-world models do not perfectly represent the true score function, leading to slight discrepancies between ideal theoretical assumptions and practical behavior.
>
> Regarding the weighting question, we tested this on Stable Diffusion and observed that higher guidance scales generally perform better, as they provide stronger correction for errors made by the diffusion model. The results are shown in the following table:
>
> |  | HPSv2 | ImageReward |
> | --- | --- | --- |
> | $w_k = \frac{h_k}{2h_{k-1}}$ | 0.245 | 0.237 |
> | $w_{\text{HiGS}}(t)$ | **0.261** | **0.392** |
>
> ### **Text-to-image improvements**
> Our focus in HiGS is to improve the overall quality of generated images across a wide range of settings, whereas CLIP score evaluates only text–image alignment. Since our quality metrics, such as HPSv2, reflect human preference, we conclude that HiGS can substantially enhance sampling quality while preserving the text-to-image alignment of various models.
>
> ### **Relation to STORM**
> Please note that Section 4.1 is used only for motivation and its results are directly verifiable by using the definition of the momentum term in STORM ($\nabla f_t - \nabla f_{t-1}$) and the definition of the energy function ($p_t = \frac{1}{Z} \exp(E_t)$). We would be happy to provide further clarification or results if this does not fully address the reviewer’s question.
>
> ### **Assumptions of smoothness and bounded derivatives**
> These assumptions are standard in neural ODE papers and theoretical optimization methods. We believe these assumptions are not limiting in practice, and we have practically verified that HiGS indeed provides higher-quality samples under lower NFEs and various guidance scales.

---

> > ### Comment · Reviewer_zxKC · 2025-11-27
> >
> > Thank you for the responses!

---

### Official Review · Reviewer_BPak · 2025-11-02

**Soundness:** 3
**Presentation:** 3
**Contribution:** 3
**Rating:** 6
**Confidence:** 4

**Summary:**

This paper proposes History-Guided Sampling (HiGS), a training-free method that improves diffusion model sampling quality by leveraging past model predictions. Inspired by momentum-based variance reduction in optimization (STORM), HiGS computes a weighted average of previous predictions and uses the difference from the current prediction as guidance. The method integrates seamlessly into existing samplers without additional forward passes, requiring only hyperparameter tuning. Extensive experiments across multiple models (SD-XL, SD3, DiT, SiT, Flux) and settings demonstrate consistent improvements, with particularly strong results in low-NFE scenarios. Using SiT-XL+REPA-E, HiGS achieves FID 1.61 on ImageNet 256×256 with just 30 steps (vs. 250 baseline), establishing new state-of-the-art for unguided generation.

**Strengths:**

1. **Strong practical value**: Training-free and truly plug-and-play (Algorithm 1), addressing a critical need for faster diffusion sampling. No additional forward passes required (Section D confirms identical 6.50 iter/sec with/without HiGS).
2. **Comprehensive experimental validation**: Extensive testing across architectures (U-Net, Transformer), model scales (675M-2.6B parameters), training paradigms (standard, distilled), and tasks (text-to-image, class-conditional). Table 6 demonstrates compatibility with multiple samplers (DDIM, DPM++, PLMS, UniPC).
3. **Consistent improvements**: Win rates > 0.75 across most settings (Table 2), with particularly strong results in low-NFE scenarios (Table 1: SiT-XL+REPA FID 12.08→4.86, -60%). Figure 5 shows benefits across all CFG scales (1.5-5.0) and NFE ranges (10-30 steps).
4. **State-of-the-art achievement**: ImageNet 256×256 unguided FID 1.61 with only 30 steps (Table 3) represents significant advancement in fast, high-quality generation.
5. **Well-motivated methodology**: Clear connection to optimization theory (Eqs. 3-4) and formal error analysis (Theorem B.1). Thorough ablation studies justify design choices (Tables 7-9, Figures 10-15).
6. **Compatibility with existing methods**: Works with distilled models (Table 4), guidance interval (Table 3 baselines), and can combine with APG (Figure 6).

**Weaknesses:**

1. **Statistical significance not established**: All experiments appear to use single runs without error bars or confidence intervals. Some improvements are small (e.g., Table 1 SD3: 27.19→26.84, only -1.3%) and may not be statistically significant. Multiple seeds and significance tests would strengthen claims, especially for marginal improvements.
2. **Hyperparameter selection guidance insufficient**: Tables 10-12 show different hyperparameters per model (w_HiGS ranges 1.0-2.5, η varies 0-1, t_min 0.3-0.4) without clear selection principles. For practitioners applying HiGS to new models, the paper lacks a practical tuning protocol or recommended defaults. Figures 12-13 show t_min sensitivity (~13% performance variation), contradicting "robust" claims.
3. **Limited theoretical-practical connection**: Theorem B.1 suggests optimal weight w_k = h_k/(2h_{k-1}), but implementation uses EMA with α=0.5-0.75. This gap is not explained. Additionally, orthogonal projection and DCT filtering lack principled justification—they appear to be heuristic solutions found through trial-and-error rather than theoretically motivated.
4. **Incomplete comparison with related work**: No direct comparison with AutoGuidance (Karras et al., 2024), which is conceptually similar but requires training. Appendix C mentions compatibility but provides no quantitative results. Comparison with other training-free fast sampling methods would contextualize the contribution better.
5. **Computational overhead not fully quantified**: While Section D reports identical iteration rates, this is for SD3 at unspecified resolution. DCT/iDCT operations have non-negligible cost at high resolutions (e.g., 1024×1024 or 2048×2048). Wall-clock time measurements across resolutions and batch sizes would be valuable.
6. **Missing failure case analysis**: Section 6 mentions "biases and limitations" but provides no concrete examples. When does HiGS not help or potentially harm? Honest reporting of failure modes would aid practical deployment decisions.

**Questions:**

1. **Hyperparameter selection**: For practitioners applying HiGS to a new model, can you provide: (a) A simple default configuration that works reasonably well across models? (b) A recommended tuning protocol (e.g., "start with w_HiGS=1.5, α=0.75, then adjust based on visual quality")? (c) Sensitivity analysis showing how much performance degrades with suboptimal choices?
2. **High-resolution overhead**: What is the computational overhead of DCT/iDCT at high resolutions (1024×1024, 2048×2048)? Can you provide wall-clock time comparisons showing actual latency impact in production settings?
3. **AutoGuidance comparison**: How does HiGS compare quantitatively with AutoGuidance (Karras et al., 2024) in terms of quality, speed, and applicability? Could HiGS and AutoGuidance be combined?
4. **Diversity metrics**: Table 1 shows slight Recall decrease for SiT-XL+REPA (0.73→0.70). Can you provide additional diversity metrics (e.g., Vendi score, cluster-based metrics) to quantify diversity preservation? Are the diversity benefits claimed in Figures 7-8 statistically validated?
5. **DCT filtering justification**: Why was DCT chosen specifically for frequency filtering? (a) Have you compared with other methods? (b) Figure 15 shows Rc=0 performs reasonably well—is DCT essential for metric improvements, or primarily for visual quality? (c) What is the quantitative impact of DCT on metrics beyond FID? (d) Why you choose frequency filtering as the solution of “unrealistic color compositions in generations” issue?
6. **Advanced momentum methods**: While STORM provides the initial motivation, recent advances in momentum-based optimization (e.g., Lion, SOAP) suggest promising extensions. Do you see these as promising future directions, and what challenges would they present for the training-free nature of HiGS?

---

> ### Author Response · Authors · 2025-11-18
>
> We thank the reviewer for the positive feedback and for recognizing the contributions of our work, including its strong practical value, comprehensive evaluation, and state-of-the-art performance. Please find our detailed responses to the comments below. We happily remain open to further discussion.
>
> ### **Question about statistical significance**
> Our quantitative results are based on a large number of individual samples (e.g., 10,000 generated images for the ImageNet experiments). With such a large sample size, we believe the performance differences between methods are meaningful. Repeating this process multiple times to compute confidence intervals would require substantial computational resources, making it infeasible within the scope of our paper.
>
> ### **Hyperparameter selection**
> We have provided in-depth ablation studies in Section E, showing that the performance of HiGS is not highly sensitive to its underlying parameters (e.g., Figure 14 and 15). The most important parameters are the scale $w_{\text{higs}}$ and $t_{\min}$, which can be selected by generating a small batch of examples and visually inspecting the results. We also note that the parameters mentioned in Section F and Algorithm 2 offer strong starting points for new models.
>
> ### **Limited theoretical-practical connection**
> Our main goal for the theorems in Appendix B is to show that a simplified version of the algorithm used in HiGS can reduce the local truncation error of the Euler solver from $O(h^2)$ to $O(h^3)$. We found that other components of HiGS further improve performance in practical settings, since real-world models do not perfectly represent the score function, leading to slight discrepancies between the ideal theoretical assumptions and practical behavior. Extending our theoretical analysis to cover these additional components of HiGS is an interesting direction for future work.
>
> ### **Comparison with autoguidance**
> Autoguidance is designed to improve sampling quality over standard CFG by providing a better guidance signal, but it does not address sampling issues that arise under lower computational budgets. Since HiGS operates at the sampler level, it is compatible with autoguidance by simply replacing the CFG input to our algorithm with the corresponding autoguidance inputs. We provide a qualitative example at [this anonymized URL](https://ibb.co/7D00Ddt). We note that under lower NFEs, HiGS can also significantly improve the quality of autoguidance outputs.
>
> ### **Computational overhead in HiGS**
> DCT can be computed in real time and on the GPU using fast algorithms such as FFT. It does not introduce any computational bottleneck, even at large resolutions. We used the standard SD3 model with its original resolution of $1024 \times 1024$, and we observed that almost all inference time is spent on the model’s forward passes. Thus, DCT does not introduce any noticeable computational overhead in HiGS, even at higher resolutions.
>
> ### **Missing failure case analysis**
> In this context, biases and limitations refer to cases where the base model cannot generate a given concept within its learned distribution (for example, a specific species it was never exposed to during training). Since HiGS is a training-free method, its improvements are inherently constrained by the representational capacity of the underlying diffusion model. Thus, if the base model fails to produce a concept, HiGS may only offer limited refinement. We would be happy to include such examples in the final version of the paper.
>
> ### **Diversity metrics**
> We use FID and Recall as our main metrics to measure diversity since it is the standard practice in the field (e.g., [1, 2]). We would be happy to consider adding additional metrics in the final version of the paper if requested.
>
> [1] Kynkäänniemi T, Aittala M, Karras T, Laine S, Aila T, Lehtinen J. Applying guidance in a limited interval improves sample and distribution quality in diffusion models. Advances in Neural Information Processing Systems. 2024 Dec 16;37:122458-83.
>
> [2] Sadat S, Buhmann J, Bradley D, Hilliges O, Weber RM. CADS: Unleashing the diversity of diffusion models through condition-annealed sampling. arXiv preprint arXiv:2310.17347. 2023 Oct 26.
>
> ### **Advanced momentum methods**
> We thank the reviewer for this suggestion. Since HiGS provides a connection between Euler sampling and stochastic optimization, we believe it opens the door for future research to integrate more advanced momentum-based methods into the sampling process of diffusion models. While our work focuses on STORM as a practical choice that improves diffusion sampling, exploring alternative optimizers remains an interesting and promising direction for future work.

---

### Meta-Review · Area_Chair_QAxu · 2026-01-02

**Summary:**

Multiple reviewers pointed out a lack of statistical significance testing, a gap between the theoretical analysis and the heuristic practical implementation, and incomplete hyperparameter sensitivity analysis. The initial critiques were largely addressed through rebuttals, and the explanations provided in the second round of discussion were found to be reasonable. Hence, I recommend acceptance, with the suggestion that the authors incorporate detailed clarifications on the theoretical-experimental links and include a transparent failure case analysis in the final manuscript.

**Reviewer Concerns:**

Reviewer BPak pointed out the lack of statistical significance testing, insufficient guidance for hyperparameter selection, and unquantified computational overhead. Furthermore, the reviewer noted an incomplete comparison with related work and with diversity metric, and a disconnect between the theoretical framework and practical implementation.The reviewer also suggested including a failure case analysis to better inform practical deployment. Following the rebuttal, I think the authors sufficiently addressed the concerns. I suggest that the authors incorporate the failure case examples into the final version to enhance the paper's practical utility.

Reviewer zxKC focused on the gap between the paper's theoretical analysis and its practical implementation. In addition, this reviewer also pointed out the lack of human evaluation or statistical significance tests. From my point of view, most of the concerns have been addressed or sufficiently explained, and this reviewer has no further comments.

Reviewer etRw had many concerns, including a lack of conceptual novelty (similar to UCGM), inconsistencies between the theoretical EDM foundation and the various frameworks used in experiments, incomplete hyperparameter sensitivity analysis, inconsistency between theory and experiments, while also characterizing the SGD/energy-based theoretical justifications as "tenuous". After first round of rebuttal, this reviewer still had concerns about some of the aforementioned points. After reading the second round of rebuttal, I think the explanation is reasonable; however, I strongly suggest that the final manuscript should include detailed clarifications on these theoretical-experimental links and results in Table 1.

Reviewer 2itd asked for clearer method presentation, and pointed out that tuning may be a burden. The authors addressed these concerns.

**Reviewer Scores:**

I think all reviewers would keep the original score.

---

### Decision · Program_Chairs · 2026-01-26

Accept (Poster)